# The Role of Caspase-2 in Regulating Cell Fate

**DOI:** 10.3390/cells9051259

**Published:** 2020-05-19

**Authors:** Vasanthy Vigneswara, Zubair Ahmed

**Affiliations:** Neuroscience and Ophthalmology, Institute of Inflammation and Ageing, University of Birmingham, Birmingham B15 2TT, UK; v.vigneswara@bham.ac.uk

**Keywords:** caspase-2, procaspase, apoptosis, splice variants, activation, intrinsic, extrinsic, neurons

## Abstract

Caspase-2 is the most evolutionarily conserved member of the mammalian caspase family and has been implicated in both apoptotic and non-apoptotic signaling pathways, including tumor suppression, cell cycle regulation, and DNA repair. A myriad of signaling molecules is associated with the tight regulation of caspase-2 to mediate multiple cellular processes far beyond apoptotic cell death. This review provides a comprehensive overview of the literature pertaining to possible sophisticated molecular mechanisms underlying the multifaceted process of caspase-2 activation and to highlight its interplay between factors that promote or suppress apoptosis in a complicated regulatory network that determines the fate of a cell from its birth and throughout its life.

## 1. Introduction

Apoptosis, or programmed cell death (PCD), plays a pivotal role during embryonic development through to adulthood in multi-cellular organisms to eliminate excessive and potentially compromised cells under physiological conditions to maintain cellular homeostasis [1]. However, dysregulation of the apoptotic signaling pathway is implicated in a variety of pathological conditions. For example, excessive apoptosis can lead to neurodegenerative diseases such as Alzheimer’s and Parkinson’s disease, whilst insufficient apoptosis results in cancer and autoimmune disorders [2,3]. Apoptosis is mediated by two well-known classical signaling pathways, namely the extrinsic or death receptor-dependent pathway and the intrinsic or mitochondria-dependent pathway. These two pathways can be activated independently or in combination, in response to either physiologically- or experimentally-induced death stimuli. The extrinsic apoptotic pathway is triggered by various extracellular death signals, whereas the intrinsic pathway is activated by intracellular damage, oxidative stress, and deprivation of growth factors [4,5,6]. Caspases, a family of cysteine aspartate proteases, orchestrate apoptotic activities in diverse organisms, including nematodes and mammals. [7,8,9]. Thus, the intracellular molecular mechanism underlying apoptosis is evolutionarily conserved across species.

Based on the structural characteristics and the functional hierarchy, caspases are generally grouped into 3 categories: Group I are involved in inflammation, Group II are initiator caspases, whilst Group III are executioner caspases (Figure 1). Caspases involved in PCD are categorized into two distinct groups: the long prodomain containing initiator caspases including caspases-2, -8 and -9, and the short prodomain possessing executioner or effector caspases-3, -6 and -7. The initiator or apical caspases act upstream of effector caspases, either directly or indirectly activated by cleavage with aspartate substrate specificity, and then the activated executioner caspases, in turn, cleave their respective cellular substrates to orderly demise compromised cells [5,10]. Among all the caspases, caspase-2, the second mammalian caspase to be characterized, is the most evolutionarily conserved caspase [11,12], suggesting that caspase-2 may play pivotal roles in maintaining the molecular and cellular integrity of an organism during development and throughout their lives.

Despite its early discovery, retention, and conservation of sequence homology during evolution, there has been a traditional concept that caspase-2, formerly known as ICH-1 (pro-interleukin converting enzyme homolog-1 [13], is a functionally redundant initiator caspase. The gene for caspase-2 was initially identified during hybridization screening for mouse genes, which were highly expressed in neural precursor cells, and initially termed Nedd2 (neuronally expressed developmentally downregulated gene 2), as their respective mRNA levels weredownregulated during the development of the brain [14,15,16]. Nedd2 was subsequently shown to be homologous to the *C. elegans* death gene-3 (CED-3). Developmental downregulation of caspase-2 in the adult brain, lack of an explicit phenotype in caspase-2 null mice [17,18], the failure of the identification of comparatively more substrates of caspase-2 [19,20], and the inadequate new technologies to investigate its distinct activation pathways to delineate its apoptotic and non-apoptotic functions [21] are major factors that have hampered the identification of a clear functional role for caspase-2. As a result, caspase-3 has received considerably more attention than other caspases, owing to its inherently high abundance and catalytic efficiency [22,23].

However, caspase-2 has functional complexity and a much broader context than initially expected. These studies have implicated the context-dependent apoptotic function of caspase-2 in various cell death paradigms and its novel and previously unidentified non-apoptotic functions [19,24,25,26]. In line with the recent data, a previous study on caspase-2 already showed that caspase-2 has both positive and negative regulatory functions in apoptosis depending on the cell type, state of growth, and death stimuli [17]. Hence, current and future research studies must take into account the implications of therapeutic inhibition of caspase-2 activity to inhibit cell death upon other non-apoptotic functions of caspase-2. Here, we review the abundant literature on caspase-2 and detail how the structure of caspase-2 gives rise to its unique processing and activation and the myriad of caspase activation mechanisms. We then review its subcellular localization, which is linked to its activity and its role in developmental pathways. Finally, we review its role in the intrinsic and extrinsic pathways of caspase activation and in other physiological functions. Throughout this review, it is apparent that caspase-2 is involved in a range of diverse functions that are both apoptotic and non-apoptotic. This is further complicated by the interaction of caspase-2 with a range of adaptor molecules, dependent on the stimuli and the context in which caspase-2 is activated, thus making the decision of cell fate highly complicated.

## 2. Caspase-2 Splice Variants

The generation of two functionally distinct splice-variants of cleaved caspase-2, pro-apoptotic caspase-2L, and anti-apoptotic caspase-2S, from the same gene via alternative splicing occurs in response to pro-apoptotic stimuli [13,14] and is regulated by reversible phosphorylation on serine residues [27,28]. The study also reported that the ratio of caspase-2S to caspase-2L increased in a time-dependent manner. Endogenous ceramide generation and subsequent phosphatase activation during apoptosis are key steps in the alternative splicing of caspase-2 mRNA, a link between the signal transduction pathway and alternative splicing. The overexpression of the long isoform caspase-2L induces cell death, whereas its short isoform caspase-2S attenuates caspase-2 activation and eventually cell death, indicating that it acts as an endogenous inhibitor of apoptosis involving pro-survival activities, including DNA repair [28,29,30,31]. In support of these observations, the two splice variants of caspase-2 mRNA transcripts are expressed in rat hippocampus after global cerebral ischemia, and both forms in humans and mice share high sequences homology [32]. The upregulation of nucleotide excision repair factor (xeroderma pigmentosum, complementation group C (XPC)), a critical DNA damage recognition factor, downregulates anti-apoptotic short isoform caspase-2S in response to DNA damage [33]. The anti-apoptotic caspase-2S is short-lived and hence not normally expressed during neuronal development and/or expressed at low levels under certain stress conditions depending on cell types [24,34,35]. It is possible that caspase-2S functions in cell cycle and DNA repair upon DNA damage. Collectively, these observations indicated that the critical role of caspase-2 activities is both pro- and anti-apoptotic.

## 3. Unique Structural Features of Caspase-2 in Relation to Its Activation and Processing

Despite all the discrepancies, accumulating evidence indicates that activation and processing of caspase-2 occur rapidly in response to both extrinsic and intrinsic apoptotic signaling pathways or independently of these two classical cell death pathways [30,33,36,37,38,39,40,41]. Hence, it is worthwhile considering its unique structural features and various activation mechanisms for a comprehensive understanding of how caspase-2 can function either as an initiator or an executioner caspase.

Based on the structural properties, caspase-2 is considered as an initiator or apical caspase, those that possess long amino-terminal prodomains termed as caspase recruitment domains (CARDs) and C-terminal catalytic domain containing large and small subunits (Figure 2). CARD is considered as a protein–protein interaction domain that enables these proteases to recruit and assemble with adaptor proteins to form caspase activation complexes and to initiate their activation. All caspases are synthesized in cells as catalytically inactive monomeric zymogens (precursors) or pro-enzymes and require a highly and orderly regulated cascade of activation processes to initiate the apoptotic signaling pathway [42,43,44].

There are at least three distinct caspase activation mechanisms that have been identified in mammals: (1) recruitment activation—zymogens are recruited into an oligomeric activating complex by their CARD prodomain (Figure 2B); (2) autoactivation—caspases initiate their own activating cleavage; (3) transactivation—procaspases are activated by another caspase [44,45,46,47,48,49]. Although caspase-2 contains the hallmarks of an apical caspase, it can be activated by different mechanisms in a context-dependent manner. How caspase-2 is activated via these activation pathways will be reviewed later.

The formation of fully activated initiator caspases, including caspase-2 from their mature zymogens, result from their homodimerization and subsequent proteolytic cleavage between domains, such as the removal of the prodomain from the large subunit and the separation of large and small subunits in the short catalytic domain [45]. Consequently, both large and small cleaved fragments (a heterodimer of a zymogen comprises newly cleaved small and large subunits) from two procaspase molecules are assembled to form active heterotetramers (Figure 3), although, in the inter-subunit, they are cleaved after dimerization. Cleavage of the linker region, however, is not required for the catalytic activity of caspases since the functional active site is generated without its cleavage upon substrate binding [50,51]. Thus, activation of caspase-2 is a two-stage process, yielding fully active caspase with enhanced catalytic-activity to ensure fast and irreversible cell death [7,52,53,54,55,56,57]. However, it is not clear whether processing/cleavage of caspase-2 by other caspases act as an amplification mechanism of death signaling events or represents a specific/cleavage mechanisms (transactivation) where caspase-2 zymogens cannot be processed by auto-catalysis due to the existence of its several isoforms. To resolve this issue, further structural studies are required in caspase-2-mediated apoptotic pathways.

Although cleavage of both initiator and effector caspase can be used as an indicator of complete activation and activity, initiator caspases do not require cleavage for their activation [58]. Hence, we argue that this concept is only applicable to certain initiators, for example, caspase-8 and -9 [59,60,61] since cleavage is a critical part of the activation process followed by dimerization for most of the initiator caspases. Despite interdomain processing of caspase-9 differing markedly from that of caspase-8, the activation of these caspases is independent of prodomain cleavage [49,62,63]. However, in contrast to these observations, a recent study found that both dimerization and coordinated cleavage of the caspase 8 zymogen are necessary for its efficient activation and to trigger apoptosis both in vitro and in vivo [64].

Unlike upstream caspases, the activation mechanisms of downstream executioner caspases with short prodomains are relatively simple as they do not require adaptor proteins to form caspase-activating complexes or require oligomerization for their activation. Rather their precursors exist constitutively as stable homodimers in solution with an unproductive conformation in their active site and are readily activated by apical caspase-induced interdomain proteolytic cleavage [42,65,66,67]. Hence the activation process of effector caspases is a direct consequence of the activation of a specific initiator caspase to amplify the cascade of activation. It is not well understood why the precursors of initiator caspases exist as monomers while effector caspases exist as dimers at physiological concentrations. However, it has been suggested that the nature of both caspases are partly because of the interaction of their weak and strong hydrophobic states of the dimer interface in initiator and effector caspases, respectively [49]. Long prodomain caspases normally exist as monomers in solution at physiological concentrations, but caspase-2 is found as a dimer of the p19 and p12 subunits and the unique homo-dimerization of caspase-2 is driven by ligand binding and stabilized by a disulfide covalent bridge at the dimer interface [51,68,69]. Mutations of the relevant Cys residue at the dimeric interface did not affect the ability of recombinant caspase-2 to dimerize or to undergo auto-catalytic cleavage [53,63], suggesting that inter-subunit cleavage may occur by an unknown intermolecular mechanism. Hence further structural and biochemical studies are required to analyze the dimer interface stability both under in vitro and in vivo conditions.

Despite these studies indicating that caspase-2 exists as a dimer under physiological conditions, why is caspase-2 activated through various distinct and complex activation mechanisms? This is the most compelling question about its mode of activation. The overexpression or upregulation of caspases above their oligomerization threshold level stimulates their auto-activation independently of known activation platforms, but when their level of expression and catalytic activity are below the normal physiological state, the recruitment activation mechanism is in place to attenuate their catalytic activation processes [35,44,49,51,53].

The complexity of environmental stresses at both cellular and organismal levels is also a strong selective force to choose the different caspase activating complexes in a context-dependent manner [70]. Despite the intense investigation of different activating pathways for caspases, all of these pathways function under a common “proximity-induced” model system since both proximity-driven dimerization and induced conformational models are two strands of this model. The proximity-driven dimerization model differs from the induced conformational model since the former emphasizes the dimerization process, whereas the latter relies on the orientation of active site conformation [42,63,71]. Hence, in this review, it is worthwhile considering the existing mechanisms by which caspase-2 is activated during different apoptotic paradigms, both in in vitro and in vivo conditions.

### 3.1. Caspase-Recruitment Activation Complex-Dependent Proximity-Induced Dimerization and Conformation Models

Despite the incomplete current understanding of how apical caspases are activated, the CARD-regulated recruitment activation mechanism appears to be important for the caspase zymogens with low levels of endogenous catalytic activity under physiological conditions. In this mechanism, the high molecular weight caspase-activating complexes function as platforms to increase the local concentrations of these zymogens to facilitate their activation via both induced-proximity dimerization and conformational changes in response to various death stimuli [44]. For example, the well-known apical caspase activating complexes include the apoptosome, which allosterically upregulates caspase-9 activation and the death-inducing silencing complex (DISC), which then activates both caspase-8 and -10 [43,46,72,73,74,75,76]. Similar to other initiator family members, caspase-2 sequesters into an oligomeric activating complex such as the PIDDosome, containing the p53-induced protein with a death domain (PIDD), receptor-interacting protein-associated Ich-1/Ced-3-homologue protein with a death domain (RAIDD), and caspase-2 and DISC in response to various cell death signals [42,69,77,78,79,80]. However, it is also worth considering how activation-complexes/platforms facilitate the activation of these initiators and whether this activation mechanism is a primary event for the induction of apoptotic or non-apoptotic functions. At present, the comprehensive understanding is that the oligomeric complexes provide proximity-induced dimerization and orient them to undergo conformational changes in their active sites necessary for their complete activation either directly through their interaction with caspase zymogens or indirectly by facilitating their oligomerization [42,63,71,81,82].

The key event that triggers the activation of apical mature zymogens after dimerization is the limited auto-cleavage that is required to generate the active site, which is indispensable for their proteolytic activity [5,64]. Upon dimerization, all caspase zymogens form two active sites (one active site/monomer), and these sites are configured by five loops in which four loops (L1, L2, L3, and L4) are derived from one monomer and a 5th loop L2′ from an adjacent monomer (this loop is indispensable for stabilizing the “activated” conformation of the active site through intimate interactions with loops L2 and L4). The L2′ loop of zymogens exists in an unproductive (closed) conformation, and the activation cleavage allows the L2′ loop to adopt the productive (open) conformation. Hence the rearrangement of these loops facilitates the conformational changes associated with the generation of an intact active-site cleft, and these processes are unique to each caspase and are evolutionarily conserved [46,50,66,67,83,84]. A structural study of caspase-2 reports that its zymogens are held in a non-substrate binding conformation (substrate binding loops are flexible and unstructured) by a salt bridge involving Arg-378 and the active conformational changes occur after the breakage of this bridge upon substrate binding [85].

Nonetheless, caspase-2 zymogens do not require auto-cleavage for the initial acquisition of activity; the inter-subunit cleavage is essential to generate threshold levels of activity and trigger the apoptotic signaling cascade [52]. Under some circumstances, recruitment of caspase-2 zymogens to the activating complexes and/or dimerization and the subsequent cleavage by other caspases, including caspase-3 and -9 is sufficient to enhance their catalytic activity [54,57,86,87,88], indicating that unique cleavage mechanisms are in place for the complete activation of caspase-2. However, both auto-activation and trans-activation of caspase-2 by other caspases are reported and are context-dependent [26].

Although recruitment of caspase-2 to higher molecular weight activation complexes are implicated in both extrinsic and intrinsic apoptotic pathways, the number of molecules interacting with its CARD-prodomain has increased over the years, indicating its interaction with a myriad of signaling pathways and demonstrating its functional versatility [24,79,89,90]. Consistent with its initiator status, caspase-2 generates stable dimers in solution but, the strength of this unique structural feature may not be sufficient to activate threshold levels of catalytic activity, and therefore caspase-2 zymogens require specific activation platforms for their activation [24]. Nevertheless, this leads to caspase-2 activation, although it is not known whether this is indeed how this protease is activated endogenously. At present, the stoichiometry amongst PIDD, RAIDD, and caspase-2 remains controversial; most importantly, whether the PIDDosome simply facilitates its auto-catalytic cleavage or allosterically promotes its activity is also unclear [91]. In this review, we now consider past and present evidence of caspase-2 activation complexes and their identified physiological relevance in determining mammalian cell fate.

### 3.2. RAIDD- and RIP1-Mediated Activation of Caspase-2 Independently of PIDD: Implication on Apoptosis, Tumor Suppression, NF-kB, and MAPK Activation

Despite the pro-apoptotic function induced by tumor necrosis factor receptor-1 (TNFR1) via the interaction with the adaptor, Fas-associated protein with death domain (FADD), and caspase-8 to orchestrate extrinsic death receptor signaling, TNFR1-mediated signaling is also implicated in the activation of pro-survival nuclear factor kappa-light-chain-enhancer of activated B cells (NF-kB) signaling pathways [92,93,94]. TNFR1-mediated apoptosis appears to involve two sequential signaling complexes; upon the normal external death stimuli, TNFR1, the adaptor tumor TNFR1-associated death domain protein (TRADD), the kinase-receptor-interacting protein (RIP1), and TNFR-associated factor-2 (TRAF2) associate to form the initial plasma-membrane-bound complex (Complex I) to rapidly activate the pro-survival NF-κB, whereas under extreme conditions and if the initial signal (via Complex I) fails to encourage TRADD and RIP1 to associate with FADD and caspase-8 to form a cytoplasmic complex (Complex II) lacking TNFR1, death is signaled.

When NF-κB is activated by Complex I, Complex II harbors the caspase-8 inhibitor FADD-like interleukin-1β-converting enzyme-inhibitory protein (FLIP(L)) to regulate this pathway [95,96,97,98,99,100]. Consistent with a role in pro-survival, TNFR1-mediated signaling also interacts with components of the downstream NF-κB machinery, such as the inhibitor of nuclear factor kappa B (I-κB), the protein kinases I-κB kinase-α (IKKα), IKK-β, and the scaffold protein IKKγ (NEMO; NFκB essential modulator) (Figure 4) [101,102]. In addition to the TNFR–caspase-8 interaction, TNFR-mediated recruitment of caspase-2 involves both pro-survival and pro-apoptotic signaling. Although this alternative signaling pathway is not exclusive, it is worth considering the possible interaction of caspase-2 in these pathways. Surprisingly not only do caspase-2, -8, and -10 pro-domains mediate activation of NF-κB [103,104,105,106] but also promote limited activation and substrate cleavage of some executioner caspases (e.g., caspase-3) that are engaged in non-apoptotic functions, including NF-κB activation [106,107,108,109]. The engagement of caspases in apoptotic as well as non-apoptotic functions indicate that pharmacological or genetic inhibition of caspases to prevent cellular death may have great impact on the normal physiological functions of mammalian cells. Therefore, current and future research work must be directed towards vigilance of the broader physiological implications of caspases than initially thought. 

The recruitment of caspase-2 to the TNFR1 complex that is also mediated via RAIDD may be considered as an alternative signaling pathway independent of PIDD. The subcellular localization and CARD-dependent oligomerization of the death adaptor, RAIDD, and caspase-2 were detected following TNF-treatment in HeLa cells upon their co-expression, suggesting that this interaction may mediate the recruitment of caspase-2 to TNFR1 [110]. Indeed, RAIDD is an unusual bipartite cytosolic adaptor molecule and is constitutively expressed in many tissues, with tissue specific-roles in regulating apoptosis in mammalian cells [111]. This adaptor molecule is also known as caspase and RIP adaptor with death domain (CRADD) since it was found to interact with caspase-2 and RIP, a serine/threonine kinase component of the death pathway via their corresponding CARD domains and carboxy-terminal death domains, respectively [77,111,112]. Initially, this adaptor molecule was thought to recruit caspase-2 to the TNFR1 signaling complex as it was found to interact with receptor-interacting kinase-1 (RIPK-1) and TRADD in response to TNFR mediated cell death in human cell lines [77,78]. Consistently, hepatocytes from caspase-2^-/-^ mice were resistant to apoptosis after combined treatment with TNFα/actinomycin, and both pharmacological and genetic inhibition of caspase-2 attenuated TNFα-mediated mitochondrial dysfunction-mediated cell death [113]. Additionally, synergistic activation of caspase-2 and -8 is also reported in homeobox A5 (HOXA5)-induced apoptosis, which appeared to be mediated by TNF in p53-mutant breast cancer cell lines [114]. It is possible that in the absence of activating complexes, a single caspase is incapable of mediating cell death; the synergistic function of corresponding caspases may be an alternative pathway to demise compromised cells.

Despite these concerted efforts, strong evidence for the recruitment of caspase-2 at this molecular complex is lacking. The association of these molecular components was documented after overexpression of RAIDD and under strict experimental conditions while neither catalytically inactive caspase-2 nor dominantly negative RAIDD constructs failed to block TNFR1-induced cell death [17,35,77]. However, we cannot underestimate the importance of this pathway, which may be context-dependent, with some cell types having alternative apoptotic pathways based on a particular death stimulus.

Consistent with TNFR mediated pro-survival signaling, CARD-mediated interaction of caspase-2 with TRAF2 and RIP1 has also been identified. The recruitment of endogenous caspase-2 into this inducible large protein complex was shown to activate NF-κB and p38 mitogen-activated protein kinase (MAPK) independently of caspase-2 proteolytic activity, thought to involve pro-inflammatory pathways, suggesting that the association of caspase-2 to this complex is sufficient for this novel non-enzymatic function [104]. Furthermore, limited activation of caspase-2 is also supported by another study, which indicated that caspase pro-domain-mediated activation of NF-κB requires an appropriate activating complex to provoke activation and its limited proteolytic activity of a small cellular fraction of particular caspase [105,106]. Thus, caspase-2 may become activated independently without inducing apoptotic signaling. The importance of TNF-signaling complexes in terms of mechanism of activation of caspase-2 and their physiological relevance in apoptosis in mammalian cells should, therefore, be further addressed [19,53].

In addition to non-neuronal apoptosis, RAIDD-mediated caspase-2 activation was demonstrated in neurotrophic factor deprivation-induced apoptosis of PC12 cells and sympathetic neurons [115]. This study further reported that the overexpression of the caspase-2-binding adaptor protein RAIDD is positively correlated with its aggregate formation, which, in turn, functions synergistically with trophic factor deprivation to induce apoptotic cell death [115]. In agreement with this, downregulation of endogenous RAIDD by small interfering RNA (siRNA) was shown to inhibit trophic factor deprivation-induced cell death in both PC12 cells and sympathetic neurons, whereas DNA damage-induced cell death was not inhibited in this model. Hence this study suggested that RAIDD and caspase-2 interaction is associated with trophic factor deprivation-induced neuronal apoptosis, but not in DNA damage-induced neuronal cell death, which is not mediated by caspase-2. These observations further support the notion that apoptotic pathways in the same system may differ depending on the initiating stimulus, and hence adapter molecule mediated caspase-2 activation is context-dependent.

Consequently, a recent study convincingly demonstrated that PIDD-null neurons treated with Aβ (β-amyloid) or nerve growth factor (NGF) deprivation, triggered the recruitment of caspase-2 to RAIDD-mediated activating complexes and induced caspase-2-dependent neuronal cell death [53], demonstrating that the activity and function of neuronal caspase-2 depend on RAIDD, but not PIDD. This study also documents that either caspase-3 or -7 is responsible for cleavage of caspase-2 in this model. On the other hand, it is not clear whether transactivation of caspase-2 by other caspases is in place during RAIDD-mediated signaling as an amplification loop in this death pathway or whether this may be considered as another unique activation mechanism of caspase-2.

Recently, a novel RAIDD-caspase-2-mediated intrinsic cell death pathway was reported in adult T-cell leukemia/lymphoma (ATLL) following treatment with an inhibitor of deacetylase enzymes (LBH589), an anticancer drug, both in vitro and in vivo [116]. This study further claimed that despite the activation mechanism being similar to the PIDDosome pathway, siRNA-mediated knockdown of neither PIDD nor RIP1 altered LBH589-induced apoptosis. In contrast, siRNA to caspase-2, RAIDD inhibited both mitochondrial outer membrane permeabilization (MOMP) and LBH589-induced apoptosis in a PIDD-independent manner. In this scenario, RAIDD appeared to be the main mediator of caspase-2 recruitment since siRNA to RAIDD effectively suppressed the cleavage of caspase-2 and cytochrome c release without BH3 interacting-domain death agonist (Bid) cleavage, suggesting that caspase-2 acts directly on mitochondria. Indeed, the caspase-2-dependent apoptotic program can bypass a deficiency in p53 and an excess of B-cell lymphoma 2 (Bcl-2) [117]. Some anti-cancer drugs like LBH589 may act upstream of the p53 pathway and cause the degradation of mutated p53 protein via an unknown mechanism [116]. Therefore, in the absence of PIDD, the downstream adaptor proteins such as RAIDD or any other unidentified interacting molecule may be in place to act as key regulators in orchestrating apoptotic signaling pathways.

Overall, it should be further clarified whether non-apoptotic functions of caspase-2 in certain cell types require just its association with activating complexes but not cleavage, whereas, its well-known proteolytic role in apoptosis requires both its association to activating platforms and consequent proteolytic cleavage.

### 3.3. PIDD-Mediated Activation of Caspase-2

Mounting evidence reports that the most prominent physiological caspase-2 activating complex or platform is known as the “PIDDosome”. The PIDDosome associated with apoptosis is composed mainly of three molecular components: cytoplasmic protein PIDD, the bipartite adaptor protein RAIDD, and caspase-2. PIDD recruits both RAIDD and caspase-2 and are assembled to form a higher molecular weight complex of ~700 kDa in response to genotoxic stress (Figure 5) [80,112,118,119]. While the interaction between PIDD and RAIDD is mediated by a homotypic interaction between their death domains, the association of RAIDD with caspase-2 occurs via their corresponding CARD domains [112,119]. This high molecular weight caspase-2 activating platform was initially implicated in DNA damage-induced apoptotic cell death [80], and, consequently, some in vitro studies provided evidence for PIDDosome function in genotoxic-stress-induced apoptosis [120,121,122]. Intriguingly, the recent identification of novel PIDDosome complexes with different molecular makeup was found to be involved in non-apoptotic functions, including DNA repair and NF-κB-activation.

These complexities have made the molecular map connecting genotoxic stress with PIDD-caspase-2-mediated cell death more challenging. Despite PIDDosome formation being reminiscent of apoptosome and DISC, the prominent feature of these activating complexes is their tightly regulated physiological functions that contribute to the fine-tuning of the various degree of DNA-damage responses via pro-survival and pro-apoptotic signaling pathways [35,123,124,125,126]. With this in mind, the next question arises: how does PIDD respond to genotoxic stress to generate activating complexes for orchestrating entirely two different signaling pathways? In response to genotoxic stress, full-length PIDD undergoes sequential auto-proteolytic cleavage to yield PIDD-N and PIDD-C fragments. Nevertheless, under the condition of severe genotoxic stress, further interdomain cleavage of PIDD-C occur to generate PIDD-CC. Whereas recruitment of RIP1 with PIDD-C to mediate pro-survival NF-κB signaling upon mild genotoxic stress, PIDD-CC assembles with RAIDD in response to severe damage to ignite pro-apoptotic caspase-2-mediated death signaling pathway. Thus, DNA damage induces the auto-proteolysis of PIDD and appears to be a molecular switch in determining the fate of cells [123,126,127,128,129]. Nevertheless, how PIDD-caspase-2 interaction with other adaptor proteins function as a molecular switch to control the life and death pathways in various cellular paradigms remain controversial. Therefore, the following section will give an overview of current perspectives relevant to the functions of various PIDDosomes in response to a variety of death stimuli.

#### 3.3.1. Implication of PIDDosome in Cellular Apoptosis

Various independent studies have demonstrated that the formation of the higher molecular weight complex can be triggered by overexpression of PIDD, co-incubation of these trimolecular components at 37 °C in vitro (also time- and salt-dependent procedures) or spontaneously upon cell rupture [35,80,101,112,119]. Despite the ability of this activating complex to form from cell extracts under strict experimental conditions, the formation of such a complex has not been detected in extracts from apoptotic cells after DNA damage [54,130]. However, it is currently unknown whether this complex is formed under normal physiological conditions upon DNA damage and/or the technologies used to investigate their presence in such conditions are not efficient. Therefore, the formation of the PIDDosome and its physiological relevance is incomplete as caspase-2 activation has been shown in the presence of such artificial in vitro triggers and may lead to erroneous conclusions in PIDD/caspase-2 interactions in apoptosis.

The recruitment of caspase-2 to this activating complex enhances its sequential proteolytic cleavage to yield smaller molecular fragments (p31, p19, and p12 subunits), indicating that caspase-2 requires such activating platform for its activation in vitro [80,82,124]. Consistent with these observations, the upregulation of PIDD enhances caspase-2-mediated apoptosis, whereas inhibition of PIDD and its interacting partners caspase-2 and RAIDD appears to have a negative effect on the formation of the PIDDosome and the attenuation of pro-apoptotic functions in vitro [80,101,112,118,121]. However, enhanced activation of caspase-2 after overexpression of PIDD is detrimental to cells and leads to apoptosis, whereas mild activation of caspase-2 in response to modest DNA damage led to beneficial outcomes [7,124,131]. Additionally, the downregulation of PIDD expression reduced caspase-2 activation and delayed cell death in vivo [88]. It is sobering to note that the activation of caspase-2 in response to various death stimuli, including endoplasmic reticulum (ER) stress, cytoskeletal disruption, and optic nerve injury, has been well documented [132,133,134,135,136,137,138]. Only DNA damage and heat shock have been experimentally shown to involve the formation of PIDDosome or complexes that have similar molecular architecture; for example, identification of stress-induced formation of Apaf-1-independent putative caspase-2 activation complex [54,80,82,139,140]. A recent study also demonstrated that caspase-2-mediated execution of NGF deprived and Aβ treated neurons required RAIDD recruitment, not PIDD, to form the activation platform [53]. This study further claimed that although the entire PIDDosome is not necessary for caspase-2 activity in neurons, other unidentified molecular components may take part in the formation of this caspase-2 activation complex in PIDD null neurons. Hence, death stimuli that synergistically activate both PIDD and caspase-2 have been very limited or, more importantly, need further clarification whether the caspase-2 activating stimuli are biological triggers of PIDDosome [124] or any other putative signaling complex, for example, DISC [79] or other unidentified activating platforms.

DNA damage-induced cell death can be mediated in a p53 transcriptional regulatory activity-dependent and independent manner [80,82,141,142,143]. p53 transcriptional activity mediated generation of various pro-apoptotic genes, including Bax and p53 upregulated modulator of apoptosis (PUMA), have also been reported in the absence of PIDD-caspase-2 interactions in the intrinsic pathway of apoptosis [144,145,146]. Although not enough physiological evidence exists, there is a possibility of the existence of alternative pathways by which DNA damage-induced generation of PIDDosome may occur independently of p53 transcriptional regulatory functions to activate caspase-2, which in turn can function upstream of MOMP to cleave and activate Bid and provoke classical apoptotic cascade events [7,28,80]. The constitutive translocation of the nuclear pool of procaspase-2 to the cytosol appears to play a role in the cleavage of Bid in the intrinsic death pathway [30,147]. Consequently, p53-dependent activation of caspase-2 releases mitochondrial cytochrome c in response to DNA damage in engineered H1299 cell lines [121]. This study further reported that siRNA-mediated silencing of RAIDD, PIDD, and caspase-2 significantly reduced apoptosis in these cells. Moreover, the validity of these results was further enhanced by previous studies that have shown similar observations; p53-mediated activation of caspase-2 in mitochondrial apoptosis [148,149,150,151] and the attenuation of PIDD-induced apoptosis upon the antisense inhibition of PIDDosome components [80,118,120]. However, genotoxic-stress-induced caspase-2 processing and acquisition of its catalytic activity are dependent on caspase-9 [88]. Taken together, p53-dependent caspase activation and the engagement of pro-apoptotic Bcl-2 family proteins are considered as hallmark features of mitochondria-mediated apoptosis after DNA damage [118,121,128,152].

Although previous studies have highlighted the significance of the PIDDosome in activating caspase-2 under genotoxic stress, paradoxically, genetic inhibition of either PIDD or RAIDD by siRNA under in vitro conditions and phenotypes of PIDD and RAIDD deficient mouse models failed to provide further support for these conflicting observations [118,120,130,153]. Remarkably, subcellular localization, nuclear translocation, and caspase-2 activation at high molecular weight complexes were not affected by the loss of either PIDD or its adapter molecule RAIDD in PIDD^-/-^ mice after DNA damage and suggests that at least one alternative PIDDosome-independent mechanism of caspase-2 activation exists in mammals [24]. For example, DISC-mediated caspase-2 activation is an alternative mechanism to the PIDDosome activated pathway in response to DNA damage [79]. More strikingly, these studies convincingly indicated that PIDD is always not an important piece of the puzzle in the pathway of caspase-2 activation under certain conditions. Caspase-2-mediated apoptotic signaling is likely to be redundant in certain cell types and may function as an ancillary or backup mechanism of p53-dependent cell death in other cell types [7]. It is worth pointing out that p53-independent coupling of DNA damage abrogates mitochondrial function via several mechanisms, including the cytosolic translocation of the nuclear protein Nur77 and the activation of nuclear and/or cytosolic caspase-2 [28,117,154,155,156]. p53 was also found to play a role in developmental neuronal apoptosis regulated by tropomyosin receptor kinase A (TrkA) and p75 receptors, whereas in mature neurons, p53 functions as a regulatory molecular switch leading to death [157,158,159,160]. Taken together, the discrepancy concerning the PIDD-mediated physiological activation of caspase-2 remains elusive in the pro-apoptotic signaling pathway, and the PIDDosome appears to form under extremely strict conditions, e.g., after overexpression of PIDD and/or RAIDD or severe DNA damage.

#### 3.3.2. Implication of PIDDosome in Cellular Apoptosis

In spite of the fact that the precise molecular mechanisms of caspase-2 activation in tumor suppression remain unclear, growing evidence emphasizes that caspase-2 has a significant role in this aspect. Aberrant cell survival is a major obstacle in cancer treatment since tumor cells express enhanced levels of multiple DNA damage repair and cell cycle arrest-related genes, including PIDD, and thereby challenge therapeutic interventions [131]. For example, an elevated level of pro-apoptotic caspase-2L is a candidate in tumor suppression [161] and is involved in Fas-mediated cell death in human leukemic cells [29]. Consequently, downregulation of caspase-2L and upregulation of caspase-2S mRNA and levels of total caspase-2 mRNA expression has been implicated in tumor progression in vivo [31,162] and more importantly the nucleotide excision repair factor XPC has been shown to enhance DNA damage-induced cell death by downregulation of the antiapoptotic caspase-2S [33]. However, these studies failed to indicate how spliced forms of caspase-2 are activated in this pathway. The upregulation of expression or activity of splicing regulators that enhance the generation of pro-apoptotic caspase-2L might have significant therapeutic potential in cancer treatment.

In addition to caspase-2, other caspases such as caspase-6, -7, and -8 also act as tumor suppressors [163,164,165], although the genetic mechanisms underlying the activation of these caspases and their relevance to tumor suppression remain to be elucidated. More remarkably, genetic inhibition of caspase-2 by siRNA suppresses apoptosis of tumor cells induced by genotoxic insults such as etoposide, cisplatin, and 5-fluorouracil (5-FU) in tumor cells [7,120,147,166] and DNA enzyme (DZ13) induced caspase-2 activation has also been implicated in tumor suppression [122]. Consistent with these results, mouse embryonic fibroblasts derived from caspase-2 null mice also exhibited an attenuated or delayed apoptotic response following treatment with cytotoxic drugs and γ-radiation and interestingly, these cells showed high rates of proliferation than their wild-type counterparts, suggesting that caspase-2 loss mediated deregulation of the cell cycle and oncogenic transformation with E1A/RAS activation play key roles in this paradigm [7,79,167,168,169]. Nevertheless, these caspase-2 null MEFs failed to show significant responses to drugs that promote apoptosis via cytoskeletal disruption such as docetaxel and cisplatin [7,133]. Delayed death, or mitotic catastrophe, can occur in p53-deficient cells and was described as the main form of cell death induced by ionizing radiation [170].

Recent studies have shed light on p53-mediated caspase-2 activation upon tumor suppression, especially in stress and DNA damage-induced cell death models, and it is already well documented that the p53 tumor suppressor gene is indispensable for DNA damage-induced apoptosis. Although there is no direct interaction between p53 and caspase-2, there is a bidirectional functional connection between p53 and caspase-2 in the molecular map connecting the embarking stimuli of genotoxic-stress-mediated apoptosis, indicating that p53 acts as an upstream regulator of caspase-2 activity (Figure 6) [120]. Furthermore, an association of caspase-2, PIDD, and RAIDD has been detected in untreated as well as in 5-FU-treated HCT116 cells, suggesting spontaneous interaction of caspase-2, PIDD, and RAIDD also occurs under normal cell conditions [120]. The presence of PIDDosome in wild-type cells and PIDD- and RAIDD-independent 5-FU induced caspase-2 processing and cytochrome c release in this study misled the hypothesis of the function of this trimolecular complex in activating caspase-2. However, upregulation of p53 in 5-FU-treated cells after the suppression of caspase-2 proteolytic activity may be explained by the requirement of caspase-2 in the turnover of p53 [120] and/or the cellular responses to DNA damage that is usually accompanied by elevated p53 expression resulting in protein stabilization and/or transcriptional activation of genes involved in cell death and cell cycle arrest pathways [171]. Downregulation of p53/p21 regulated caspase-2 has also been detected in the presence or absence of DNA damage in human H1299 tumor cells, indicating that caspase-2 repression by p53 may be important in determining cell fate when necessary by inhibiting caspase-2-dependent cell death [121].

In support of these observations, silibinin, a natural flavonolignan was shown to activate p53 by ataxia-telangiectasia mutated (ATM)-Chk2 pathway-mediated phosphorylation, which in turn induced caspase-2-mediated cleavage of Cip1/p21 and apoptosis in bladder transitional-cell papilloma RT4 cells [172]. This study further reported that the p53 inhibitor reversed silibinin-induced caspase-2 activation and caspase-2 inhibitor reversed p53 phosphorylation, suggesting a bidirectional feedback regulation between them. The rapid translocation of p53 and Bid into mitochondria, leading to increased permeabilization of mitochondrial membrane and cytochrome c release and c-Jun N-terminal kinase (JNK)1/2 activation, acts as a connecting link for p53-mediated caspase-2 activation in this model. Consistent with these observations, UV-induced upstream caspase-2 activation and cell death were reduced in Bax-deficient cells [173]. Both positive and negative feedback loops are distinct features of the auto-regulation of the p53 expression mediated via ATM/ATM and Rad-3-related (ATR) regulation [174,175]. Collectively, it appears that caspase-2 acts upstream of mitochondria, and the ATM-Checkpoint (Chk)2 pathway-mediated phosphorylation of p53 may be associated with PIDDosome-induced apoptosis [7,126].

Cell cycle progression or DNA damage can lead to mitotic catastrophe when Chk2 is inhibited and thus can sensitize proliferating cells to chemotherapy-induced apoptosis [176]. PIDDosome expression and caspase-2 activation have also been implicated in chemotherapy-induced apoptosis in renal cell carcinomas [31]. A functional relationship between apoptotic and wild-type p53-dependent-PIDD expression was recently found in patients with oral squamous cell carcinoma, but not with mutated p53 [177,178]. Nevertheless, depletion or acute inhibition of Chk1 is sufficient to restore gamma-radiation-induced apoptosis in p53 mutant cells since an alternative apoptotic program that is not affected by p53 loss or overexpression of Bcl-2/x but requires ATM, ATR involvement, can be engaged in caspase-2-dependent apoptosis that bypasses p53 deficiency and causes overexpression of pro-survival Bcl-2 [117].

Recently, caspase-2-mediated suppression of the survivin gene, a general regulator of cell division and cytoprotection in tumor cells has been reported, and this process appears to associated with the proteolytic cleavage of the NF-κB activator, RIP1, suggesting that caspase-2 functions as an endogenous inhibitor of NFκB-dependent cell survival and contributes to suppression of tumorigenicity in vivo [179]. To confound this, a protein complex consisting of caspase-2, TRAF2, and RIP1 is found to activate NF-κB and p38 MAPK through the caspase recruitment domain of caspase-2, independently of its proteolytic activity [104]. Hence, the pro-survival pathway appears to require caspase-2 interaction independently of its proteolytic function and RIP1 cleavage, whereas caspase-2-mediated apoptosis requires both. Furthermore, the downregulation of caspase-2 was implicated in various types of cancers in patients. The anti-tumor function of caspase-2 requires catalytic site Cys-320 and site Ser-139 in mouse models, and these two residues are required for the inhibition of NF-κB activation, sustain the G2/M checkpoint and induce apoptosis, indicating combined functions of caspase-2 in the implication of its anti-tumorigenic function [180]. In addition to NF-ĸB and cell cycle arrest, inhibition of Akt and extracellular signal-regulated kinase (Erk)1/2 are also key factors in inducing apoptosis and inhibiting tumor growth in colon cancers [181,182].

Although the mechanism by which PIDD promotes cell cycle arrest and drug resistance is not well understood, the significance of mutations in the p53 tumor suppressor or overexpression of anti-apoptotic Bcl-2 family proteins is implicated in malignant transformation and therapeutic resistance. Chemotherapy resistance is one of the major problems in therapeutic interventions for cancer treatments. For example, prolonged cisplatin treatment promotes its resistance with enhanced repair capacity and suggests that PIDD act as a regulator of the chemotherapy response in human lung tumors [131]. Somewhat surprisingly, the molecular mechanisms underlying chemotherapy resistance appears to depend on PIDDosme-mediated positive feedback loop that involves inhibition of E3 ubiquitin ligase mouse double minute (Mdm)2, a cleavage target of caspase-2 and reinforces p53 stability and activity, contributing to cell survival and drug resistance (Figure 7). PIDD-induced caspase-2 directly cleaves Mdm2 at Asp 367 and results in the loss of the C-terminal of a really interesting new gene (RING) domain responsible for p53 ubiquitination. As a consequence, N-terminally truncated Mdm2 binds p53 and promotes its stability [131].

Paradoxically, PIDDosome-independent tumor suppressor function of caspase-2 has been reported. The in vivo study showed that even though neither PIDD nor caspase-2 failed to suppress lymphoma formation triggered by γ-irradiation or 3-methylcholanthrene-driven fibrosarcoma development, caspase-2 was anti-tumorigenic in response to aberrant c-Myc expression (a carcinogen forming bulky adducts with DNA), independently of PIDD, Bid or Trail and caspase-2 mediated tumor suppression was associated with reduced rates of p53 loss and increased dissemination potential of tumor cells. Nevertheless, PIDD deficiency is also associated with abnormal M-phase progression and delayed disease onset, indicating that both PIDD and caspase-2 are differentially engaged upon oncogenic stress triggered by c-Myc, leading to delayed onset of tumorigenesis [53]. Although the mechanism underlying c-Myc induced caspase-2-mediated tumor suppression remains to be elucidated, previous in vivo studies also give strong evidence for the involvement of caspase-2 in tumor suppression in Eµ-Myc mouse lymphoma model [7,168,183,184].

Despite the discovery of caspase-2-dependent tumor suppression, some controversies exist, particularly how p53 dependent and independent regulation of caspase-2 activity enable this protease to engage in entirely two different pathways. Is it really involved in apoptosis or does it leave its main death machinery track to survival pathway in a context-dependent manner? Also, it remains obscure whether 2 splice forms both caspase-2S and caspase-2L exist in the same cellular system and participate in these two pathways depending on the severity of the cellular damage or the cell types that only contain pro-apoptotic caspase-2L are more prone to undergo cell death. However, the PIDD–caspase-2 association appears to function as a safeguard against activation of the apoptotic program in the first instance, and apoptotic cell death seems to be the final outcome in response to extreme levels of death signals.

#### 3.3.3. Implication of Caspase-2 Phosphorylation and DNA-PKcs-PIDDosome in Cell Cycle Arrest and DNA Repair

Phosphorylation-mediated regulation of mammalian caspase-2 activation is found as an important process for its biological function, although this molecular mechanism remains obscure. It was reported that phosphorylation of pro-caspase-2 at S157, a highly conserved sequence in mammals, is mediated by the protein kinase CK2 (PKCK2) and the suppression of PKCK2 leads to dephosphorylation and consequent proteolytic processing that is implicated in apoptotic cell death, indicating that PKCK2 acts upstream of pro-caspase-2 to determine the cell sensitivity to induce apoptosis [19,91]. Calcium/calmodulin-dependent protein kinase (CaMKII) is also found as another regulator of caspase-2 phosphorylation at Ser 135 and modulates the metabolic state of cells, and the sustained metabolically-regulated phosphorylation of caspase-2 prevents the induction of caspase-2 mediated apoptotic pathways leading to cell survival [127,185]. Phosphorylation of caspase-2 at S164 inhibits its induced proximity oligomerization, and autocatalytic processing [186]. More importantly, mitosis-promoting kinase (Cdk1-cyclin B1) was shown to attenuate apoptosis upstream of mitochondrial cytochrome c release by phosphorylating caspase-2 within an evolutionarily conserved interdomain sequence within a distinct serine-proline motif at Ser 340 in human and S308 in Xenopus, and this region was susceptible to phosphatase 1 dephosphorylation [187]. The study also reported that caspase-2 phosphorylation is indispensable to prevent cell death during mitosis in cell cycle progression and under conditions of mitotic arrest; cdk1-cyclin B1 activity must be overcome to prevent apoptosis. Furthermore, it is thought that metabolic regulation of caspase-2 phosphorylation in Xenopus is regulated by CaMKII, whereas this process is regulated by Chk1 in mammals as these two enzymes were found to phosphorylate Cdc25, upon DNA damage and Chk1 inhibitors downregulate CaMKII with similar potency (Figure 6) [19,188,189].

Metabolic regulation of caspase-2 dephosphorylation has also been reported and implicated in cellular control of caspase-2 activities. Enzymes that catalyze the phosphorylation of caspase-2 have been identified, such as protein phosphatase-1 (PP1) and protein phosphatase-2A (PP2A), and these phosphatases have direct and indirect effects on caspase-2 dephosphorylation [127,185,190]. PP1-mediated regulation of caspase-2 phosphorylation is involved in the control of M phase entry during vertebrate mitotic division [19,185,191,192,193,194], whereas PP2 functions indirectly to control caspase-2 dephosphorylation via an unknown mechanism. However, the overexpression of PP2A increased caspase-2 activation, Bcl-2 dephosphorylation, and mitochondrial damage and eventually led to apoptosis [190]. PP2A can act upstream of mitochondria to dephosphorylate Bcl-2, which, in turn, activates biological triggers for caspase-2 dephosphorylation leading to activation of the intrinsic apoptotic pathway. These observations might explain the cellular regulatory molecular mechanism related to its activation and significance of metabolically regulated drug resistance in tumor cells. Nevertheless, it is unclear whether caspase-2 phosphorylation occurs upon its recruitment to any unknown caspase-2 activating complexes or independently in this paradigm. It is noteworthy that p53 and caspase-2 interaction and metabolically regulated caspase-2 phosphorylation have been well documented in both pro-survival and pro-apoptotic pathways. Recent studies have also highlighted the importance of p53 as an important regulator of metabolic pathways in cells through the regulation of mitochondrial integrity, nucleotide, and antioxidant responses to its repertoire of activities by its transcriptional as well as non-transcriptional activities [195,196,197,198]. Intriguingly p53 mediated caspase-2 phosphorylation has been reported recently, and this is implicated in cell cycle arrest and DNA repair. PIDD-mediated novel activating complex (DNA–PKcs–PIDDosome) containing mainly of PIDD and the DNA-dependent protein kinase catalytic subunit (DNA-PKcs), has been found to activate nuclear caspase-2 via its phosphorylation at S122 in response to DNA double-stranded breaks (DSBs). This activating platform mediated phosphorylation appears to be indispensable for caspase-2 activation and its novel non-apoptotic functions in response to DSBs. Unlike PIDDosome, this is thought to maintain G2/M cell cycle arrest and DNA repair regulated by the non-homologous end-joining pathway [7,70]. Consistent with this study, DNA-PK-mediated suppression of p53-dependent cell death and DNA repair activities in response to DNA damage has also been reported [199,200,201].

Generally, in dividing cells, DNA damage caused by mild genotoxic insults result in the activation of cell cycle checkpoints followed by DNA repair to ensure genomic stability and integrity. The cellular responses to DNA damage are mainly regulated by two distinct kinase signaling pathways: ATM through Chk2 and ATR through Chk1. However, ATM is required both for ATR–Chk1 activation and to initiate DNA repair via homologous recombination by inducing the formation of single-stranded DNA at sites of damage through nucleolytic resection in response to DSBs [202,203]. The enhanced ATM-mediated phosphorylation of p53 in response to DNA damage and the lack of mutated ATM association in the regulation of NF-κB has also been reported [204,205].

Consistent with the previous studies, ATM/ATR mediated regulation of PIDD has been implicated in both pro-survival and pro-apoptotic pathways. It seems that PIDD function as a sensor of the DSBs and ATM-mediated phosphorylation regulates the activity of various downstream molecules, including p53, Mdm2, and Chk1, to orchestrate cellular responses leading to growth arrest or apoptosis [7,19,166,206,207,208] and thereby controlling the balance between life and death of compromised cell after DNA damage. Depending on the severity of the damage, p53 activation activates either G1 or G2 arrest, mainly through transcriptional upregulation of p21 and/or 14-3-3σ [209,210,211] and alternatively, apoptosis might be initiated through the activation of pro-apoptotic genes, such as Bcl-2-associated X protein (Bax), p53 upregulated modulator of apoptosis (PUMA) or FasR as a final outcome [144,212] independently of PIDD mediated RIP1/NEMO or RAIDD/caspase-2 activation complexes [128].

Thus, PIDD-mediated NF-κB and apoptotic machinery function differentially in a sequential manner (early recruitment of RIP1-NEMO versus late recruitment of RAIDD-caspase-2) in response to genotoxic stress [123]. It is very likely that PIDD is positioned upstream of RIP1 and RAIDD in these signaling cascades. Upon genotoxic stress, the cytoplasmic PIDD is constitutively translocated to the nucleus to encounter both RIP1 and NEMO and provoke NF-κB activation-mediated transcription of anti-apoptotic genes, whereas the caspase-2, PIDD, and RAIDD- mediated activation platform formed in the cytosol to orchestrate caspase-2 dependent death pathway in response to severe DNA damage [128]. Not only in response to DNA damage, but also for heat shock, and cytoskeletal disruption, the caspase-2 activation platform PIDDosome is generated in the cytosol, but not in the nucleus [82]. Hence the reciprocal functional connection between p53 and caspase-2 is indispensable for caspase-2 in response to certain death stimuli [120]. Indeed, PIDD activation and, more importantly, caspase-2 activation are not always synonymous with the induction of cell death, and threshold levels of caspase-2 activation must be achieved before the engagement of caspase-2-dependent apoptosis [126]. Some conflicting observations indicate that although caspase-2 promotes NF-kB activation in pro-survival pathways independently of its proteolytic function/cleavage and RIP1 cleavage, caspase-2 is also found as an endogenous inhibitor of NF-kB activation-mediated cell survival by cleaving the NF-kB activator RIP1 to inhibit the transcription of NF-kB target genes [104,179,180].

Unlike PIDDosome, PIDD has been found to interact with death-domain-containing RIP1 and NEMO/IKKγ, the regulatory subunit of the cytoplasmic IκB kinase (IKK), to form another protein activating platform called NF-κB-activating kinase complex, independently of RAIDD and caspase-2, to activate NF-κB (Figure 8) in regulating DNA-repair after mild genotoxic stress to ensure cell survival [128]. Taken together, despite caspase-2 not having direct effects on NF-κB activation, it does have indirect effects on pro-survival signaling regulation since caspase-2 and p53 have been shown to control their level of expression via a unique feedback mechanism [172,174,175]. The existence of PIDD-associated to two distinct signaling platforms indicates that RAIDD and RIP1 interact mutually in an exclusive manner with PIDD to maintain the cellular integrity as a homeostatic adaptive response in accordance with the severity of genotoxic stress.

However, there are some similarities in the mode of formation of NEMO-PDDosome in NF-κB-activation and DNA–PKcs–PIDDosome in DNA repair following genotoxic stress. Firstly, both activating platforms are implicated in pro-survival pathways. Secondly, similar to ATM-mediated PIDD-dependent phosphorylation of NEMO, caspase-2 phosphorylation is mediated by DNA-PKcs in a PIDD-dependent manner. Finally, PIDD is translocated constitutively into the nucleus to recruit RIP1 and NEMO for the activation of NF-κB signaling and/or is recruited to nuclear DNA-PKcs to activate caspase-2 phosphorylation leading to G2/M cell cycle arrest and DNA repair [19]. In agreement with these observations, ATM-mediated PIDD phosphorylation on Thr788, within the death domain (DD), in the absence of Chk1 is required for RAIDD binding and caspase-2 activation since non-phosphorylatable PIDD fails to bind RAIDD or activate caspase-2 and instead associates with pro-survival RIP1 [126]. One study provided strong biochemical evidence for the ATM-mediated phosphorylation of PIDD that appears to function as a molecular switch between cell survival and caspase-2-mediated cell death [127]. Hence, the functional cross-talk/interplay between PIDD phosphorylation and caspase cleavage may explain the molecular mechanisms underlying PIDD-mediated downstream activation of caspase-2 in response to DSBs [126,213]. Confoundingly, caspase-2-mediated apoptosis was reported in Chk1 inhibited cells following replication stress due to the DSBs and deregulation of ATM/ATR activation regardless of the PIDD association [117], suggesting that caspase-2 can mediate apoptosis in a PIDD-independent manner in this model. In the absence of p53, checkpoint signaling is regulated via a combined function of both Chk1 and the p38MAPK/MK2 pathways in response to DSBs to activate pro-survival pathways [7,19,214,215].

Nevertheless, a recent study also indicates that Chk1 is essential for neuronal survival even in the absence of DNA damage, and perturbation of this pathway increases a cell’s sensitivity to naturally accumulating DNA damage [216]. Taken together, caspase-2 phosphorylation and its subsequent inhibition of proteolytic cleavage are indispensable for cellular systems to regulate survival. These controversial observations lead to raise the following questions: Is Chk1 repression vital for PIDD-dependent and independent caspase-2-mediated apoptosis in response to DSBs? Or does inhibition of Chk1 upregulate any unidentified biological triggers in the absence of PIDD? Hence the functional link between Chk1 inhibition and caspase-2 activation needs further clarification to reconcile these conflicts. Taken together, the PIDD-caspase-2 signaling pathway cannot be overlooked in apoptotic cell death. PIDDosome mediated activation of caspase-2 is mostly implicated in its non-apoptotic novel functions compared to its well-known pro-apoptotic function. The precise role of PIDDosome-mediated activation of caspase-2 in death signaling pathway, therefore, remains obscure. PIDD appears to act as an integrator or molecular switch to control the balance between survival and death in response to DNA damage depending on the type and level of damage to cellular genomes [28,126,128].

### 3.4. DISC Mediated Activation

In addition to the role of PIDDosome in caspase-2 activation in mitochondrial-dependent apoptosis, CD95 (Fas/APO-1) death-inducing signaling complex (DISC) has also been implicated in extrinsic death signaling pathways [89]. DISC is generally composed of CD95 and TNF-related apoptosis-inducing ligand receptor 1/2 (TRAIL-R1/R2), FADD, and pro-caspase-8 or -10, in which FADD plays an important role in recruiting these caspase zymogens to the signaling complex [217,218,219,220]. The recruitment of pro-caspase-8 at the DISC results in a series of auto-proteolytic cleavage events and consequent cleavage of downstream effectors, including caspase-3, to execute the cascade of apoptotic events [43]. There are two CD95-mediated signaling pathways that exist, depending on the levels of CD95-mediated DISC formation: Type I cells with high levels of DISC formation and increased activation of caspase-8 and Type II cells with lower levels of DISC formation and activated caspase-8. In Type II cells, an additional amplification signaling loop is involved in provoking caspase-8-mediated Bid cleavage to release cytochrome c from mitochondria, leading to cell death [221,222,223,224,225,226].

In addition to caspase-8, the involvement of caspase-2 was also reported in death receptor-regulated apoptosis, but its cleavage was mediated by caspase-3 [72,218,227]. However, the molecular mechanism underlying the death receptor-mediated activation of caspase-2 in response to extrinsic death stimuli is not well understood. Interestingly, a study demonstrated caspase-2 activation and its recruitment to the DISC in response to CD95 stimulation in human T- and B-cell lines [89]. This study further claimed that despite its DISC-mediated activation, caspase-2 failed to initiate cell death in the absence of caspase-8, and thus caspase-2 activation may be an amplification loop to cleave Bid or alternatively activate NF-κB signaling pathway. In contradiction to these results, caspase-2 was shown to activate caspase-8 and is required for Bid cleavage to mediate TRAIL-mediate apoptosis [38,228] and that the pro-apoptotic caspase-2L was found to associate with activated caspase-8 at the DISC and Fas-mediated apoptotic cell death of human leukemic cells [29]. In agreement with this, other independent studies have also reported that caspase-2 mediated caspase-8 activation is indispensable for TNF-induced apoptosis and in TRAIL-induced apoptosis [91,114].

Although caspase-2 activation was implicated in CD95-mediated apoptosis [29], these observations might be explained by the interaction of CD95 and with other signaling pathways such as p53-mediated-NF-κB activation [229,230]. More remarkably, CD95-mediated DISC formation has been reported after NF-κB activation, suggesting that dynamics within this activation complex can determine life and death of cells [231], indicating that CD95 stimulation does not only lead to cell death but also in NF-κB activation. Furthermore, p53- and CD95-associated cell death has been implicated in several neurodegenerative diseases such as Parkinson’s disease (PD), amyotrophic lateral sclerosis (ALS), Down’s syndrome (DS), Alzheimer’s disease (AD) and in response to DNA damage [232,233,234,235]. In addition, the expression of apoptosis-related proteins such as RAIDD, ZIP kinase, Bim/BOD, p21, Bax, Bcl-2, and NF-κB was reported in brains of patients with DS [236].

Recent studies have shed light on the importance of caspase-2 activation in DNA damage-induced cell death. Caspase-2 is required for DNA damage-induced expression of the cyclin-dependent kinase inhibitor 1 (CDK) inhibitor p21 (WAF1/CIP1) which acts as an inhibitor of apoptosis in a number of systems to mediate tumor-suppressive functions [237,238], whereas PIDDosome independent and dependent caspase-2 activation was found to be important for DNA-damage-induced apoptosis [24,79,80]. CD95-mediated DISC formation in response to DNA damage is also an alternative PIDDosome-independent activation platform for caspase-2 [79]. This study further reported that the DISC is generated upon the p53-dependent upregulation of CD95 activation and the recruitment of caspase-8 to this complex is required for activation of caspase-2, both acting simultaneously upstream of mitochondrial cytochrome c release. In support of this, a previous study also indicated the direct interaction of both caspase-2 and -8 since their cleavage kinetics were identical on strong and weak CD95 stimulation [89]. However, this study also failed to identify the appropriate adaptor molecules that facilitate caspase-2 recruitment at DISC, and hence the mechanism underlying the caspase-2 recruitment and activation in this model is still questionable. Nonetheless, it is apparent that caspase-2 can mediate both pro-survival and pro-apoptotic signaling in response to DNA damage, to maintain genomic stability and integrity.

### 3.5. Caspase-Recruitment Activation Complex Independent Dimerization Model

Interestingly, accumulating evidence indicates that when caspase-2 is overexpressed or present at high concentrations, it can be auto-activated by its CARD-mediated homodimerization or oligomerization independently of caspase-recruitment activation complexes both in vitro and in vivo conditions [7,54,68,239]. Nonetheless, this activation mechanism is highly applicable for in vitro conditions since under experimental conditions, the proximity can be obtained by the high concentration of the caspase zymogens unless the physiological expression of caspase-2 is altered experimentally. Subsequent induced-proximity of caspase zymogens facilitates their auto-processing when brought into close proximity of each other [43]. Under physiological concentrations, caspase-2 is also constitutively dimeric in solution, and this unique feature is stabilized by inter-subunit disulfide bridge at the dimer interface, which covalently links the two monomers [68,69]. In this scenario, it is also possible that the dimerized caspase-2 in the context of the induced conformation exhibits at a perturbed interface that may greatly facilitate its catalytic activity.

In support of this, Earnshaw and his colleagues [45] reported that increasing the local concentration and/or the inducing conformational changes of initiator zymogens, can create a favorable microenvironment in which these zymogens undergo efficient trans-catalysis, the cleavage of one zymogen molecule by another. Hence, it may not be entirely surprising that caspase-2 can be activated neither upon recruitment to adaptor protein complexes nor its substrate binding. Nevertheless, both the prodomain and the carboxyl-terminal are indispensable for caspase-2 dimerization and the subsequent auto-processing under in vivo conditions [68,240]. Paradoxically, an in vitro study reported that recombinant caspase-2 lacking the CARD prodomain can be activated via auto-processing when expressed at high concentrations and suggested that disulfide bonds observed in the crystal structure are dispensable for its dimerization [52]. Although, it has been reported that this proteolytically active engineered enzyme might not require prodomain-induced dimerization or oligomerization for their activation [65]. This issue leads to the next query: is the activation of caspase-2 generally caused by increased local concentration induced-conformational changes in response to various death stimuli?

When caspase-2 is expressed at a concentration sufficient for its activation, dimerization might be the sole mechanism to orient the active site conformation to enhance its efficient substrate binding and proteolytic activities. No matter how caspases are activated, the correct formation of active sites is indispensable for their catalytic activity [46,64]. The unique features of caspase-2 reconcile the apparently conflicting opinions or views about its activation in the absence of any accessory or adaptor proteins since its activation is of fundamental importance in caspase-2-induced cell death commitment during certain death paradigms. Moreover, it is worth considering that not only caspase-2 but also the downstream executioner caspases undergo efficient auto-processing and activation in vitro by induced proximity when overexpressed or their local concentration is increased, [45,241,242]. For instance, during chemically induced apoptosis, caspase-3 is auto-processed and subsequently activated through the FK506-binding proteins and arginine–glycine–aspartate (RGD) motif rather than activated by the usual initiator caspase under physiological conditions [243,244]. In this respect, we can argue that being a dimer in solution, caspase-2 may act as an executioner caspase during certain cell death paradigms.

Surprisingly, there is no evidence so far to convincingly show that activation of caspase-2 is strictly and completely dependent on specific activating complexes and therefore the self-activation mechanism is equally possible under the conditions of increased local concentrations and/or with required post-translational modification to prompt catalytic activity of caspase-2 [24]. Furthermore, despite caspase-2 having structural similarities of initiator caspases, there are subtle differences in its activation mechanisms. Unlike other initiator caspases, procaspase-2 can oligomerize and be activated in a concentration-dependent manner without the need for their respective adaptor proteins, as opposed to caspase-8 and caspase-9, which require FADD and apoptosis protease-activating factor (APAF)-1, respectively [239]. How is it possible under physiological conditions? In this respect, caspase-2 may show its versatility to undergo different activation mechanisms presumably due to its endogenous dimeric state and its intrinsic ability to be auto-activated independently of activation complexes. Collectively, all these experimental data do not accurately recapitulate how caspase-2 is activated under physiological conditions. Hence, further structural and biochemical studies are required in terms of its unique structural features and its interactive molecular partners to address this issue.

### 3.6. Transactivation

Transactivation is also one of the most well-established classical activation mechanisms by which one caspase can be activated by another caspase. Generally, apoptotic stimuli lead to activation of upstream initiator caspases, which in turn can induce intracellular signaling pathways involving proteolytic activation of downstream effector caspases to ignite the cascade of apoptotic events [44,45,245,246]. However, intracellular proteolytic caspase-signaling pathways operate in a network-like fashion, in which initial activation of one caspase can lead to activation of multiple other family members, resulting in cleavage of many different substrates, indicating the existence of a hierarchic proteolytic pro-caspase activation network. The fact that caspase-2 does not have a direct effect on processing any of the pro-caspases except its own precursors during auto-processing indicates its distinct features as a direct effector to amplify apoptotic signaling cues; instead, caspase-2 was found to cleaved by caspase-8, -3, -9 and to a lesser extent by caspase-7 [30,55,247].

Furthermore, caspase-2 undergoes autocatalytic processing when activated and is also cleaved by caspase-3 downstream of MOMP [57]. A conflicting observation indicated that caspase-mediated cleavage does not activate caspase-2 [52]. It appears that transactivation may be an implicated mechanism for the complete and irreversible activation of cellular caspase-2 in this context and cleavage is not a proximal step of activation, but rather the final step of the activation process. Hence, it is not clear whether cleavage and activation of caspase-2 by other caspases are necessary or that this step may simply be an amplifying loop rather than an effector function for this protease in the apoptotic cascades.

### 3.7. Caspase-2 Activation and Activity

The activation and activity are two different distinct processes indispensable for caspase-mediated physiological functions [25]. Whereas dimerization and the subsequent proteolytic cleavage are required to enhance caspase activity, especially for apoptotic functions, partial activation of caspase-2 upon dimerization and recruitment at activating platforms is sufficient for this caspase to become involved in non-apoptotic functions. No matter how caspases are activated, the enhanced catalytic activity must be correlated with altered active site conformation [91] and the proteolytic cleavage of endogenous caspase-2 (51 kDa) to form three fragments of 32–33 and 14 kDa, which are then further processed into 18- and 12-kDa active subunits that are the hallmarks of its activity in apoptosis [36,72]. It is now apparent that initiator caspases do not require cleavage for their initial activation that occurs upon dimerization, and remarkably, caspase can be activated with their activity suppressed [25,53,58,130] with caspase production, processing, and activity being regulated at several different levels [45].

One of the major reasons for the paucity of knowledge on the physiological activation of caspase-2 is the unavailability of the appropriate techniques to measure the level and state of physiological activation of this caspase [82]. However, synthetic peptide substrates, detection of cleavage fragments, and unbiased affinity-ligand capture techniques (using biotin-VAD-FMK) are the most widely used methods to detect caspase-2 activity, but among them, the affinity-ligand capture technique is the best current method available to trap active caspase-2 [6,25,37,53]. Measuring caspase-2 activity by synthetic peptide substrates using VDVAD that is recognized and cleaved by caspase-2 and/or relevant inhibitors did not provide any specific information about caspase-2 substrates and its activity since this sequence motif was also found to be on substrates for other executioner caspases such as caspase-3 and -7, inadvertently inhibiting their activities [5,248,249].

Moreover, the crystal structural and biochemical studies of caspase-2 in complex with specific inhibitors reported that caspase-2 uniquely prefers a penta-peptide (VDVAD) rather than a tetra-peptide for efficient cleavage and the disruption of a non-conserved salt bridge between Glu217 and the invariant Arg378 may be important for the activation of caspase-2 [85]. Hence, it appears that phosphorylation of serine residue at different positions in the conserved region [185,190] and the cleavage occurring at the non-conserved region of caspase-2 may explain how caspase-2 can be regulated to mediate both pro- and anti-apoptotic functions.

Intriguingly, a more recent study using N-terminal combined fractional diagonal chromatography (COFRADIC) demonstrated that activated human caspase-2 shares remarkably overlapping protease specificity with the prototype apoptotic executioner caspases-3 and -7, suggesting that caspase-2 could function as a pro-apoptotic caspase once released from the activating complex. Substrate analysis of 68 cleavage sites identified in 61 proteins revealed that the protease specificities of human caspase-2, -3 and -7 largely overlap, indicating DEVD↓G consensus cleavage [36]. Nevertheless, this study further reported that specific functions of caspase-2 might not rely on its intrinsic specificity features, but rather on the context in which it is activated in response to various stimuli.

Unfortunately, the apparent functional ambiguity, multiple undefined molecular interacting partners, and lack of an overt phenotype observed in caspase-2^-/-^ mice have rendered caspase-2 long masked for its important and distinct physiological functions in the past [24]. For example, the putative caspase-2 substrates, with the exception of Golgin 160 [250], were also shown to be avidly cleaved by both activated caspase-8 and -3, rendering caspase-2 function by substrate identification to be difficult [35,79,251], indicating the potentially devastating impact on inadvertent caspase-2’s true physiological functions. However, recent studies have started shedding light on caspase-2, broadening the understanding of caspase-2 activities and substrate specificities using new technologies, such as novel caspase-2 specific probes, to resolve these problems and to explore its true and wider physiological functions.

## 4. Sub-Cellular Localization of Caspase-2

The subcellular distribution of caspase-2 may be closely related to cleavage of its substrates or association with specific interacting partners to mediate several physiological functions in different cellular compartments. However, only a few target proteins such as golgin 160, αII-spectrin, Bid, Mdm2, Huntingtin, and PARP (Table 1) have been identified as substrates for caspase-2 among more than 250 currently identified to be cleaved by other caspases [28,250,252]. Endogenous caspase-2 resides throughout the cell but, more specifically, located in Golgi, endoplasmic reticulum (ER), mitochondria, nucleus, and cytoplasm [87,250,253,254,255]. Constitutive nuclear translocation of caspase-2 from cytosol and its presence in the Golgi complex is considered as one of the most distinguishing features of this protease [24,28,256].

The unusual importin mediated bipartite nuclear import of pro-caspase-2 is regulated by two nuclear localization signals (NLS) located in prodomain, indicating that nuclear localization of caspase-2 may be indispensable for its activation prior to cell death [257,258]. Further, Paroni and his co-workers [256] also reported that caspase-2 is retained in the nucleus until the late stage of apoptosis, although the capacity of caspase-2 nuclear translocation may not correlate with its potency to mediate cell death [24,257]. In most cell types, even after the induction of apoptosis, caspase-2 is detected in the nucleus, and cytoplasmic translocation occurs during the late stages of apoptosis [24,65,215,259].

On the other hand, localization of caspase-2 in the nucleus is likely to cleave nuclear substrates, particularly in response to genotoxic stress signals [52]. Despite the need of the long pro-domain mediated caspase-2 nuclear translocation for activation, executioner caspases with short pro-domains such as caspase-3 and -7 were shown to translocate to different cellular compartments upon activation to execute apoptosis, reporting that their short domains were fully competent to direct nuclear localization, e.g., a chromosome region maintenance 1 (CRM1)-independent nuclear export signal (NES) in caspase-3 small subunit [84,249,255]. Conversely, previous studies reported caspase-2 prodomain-mediated nuclear transport and enhanced auto-activation of caspase-3 in transfected mammalian cells [240,260].

The importance of caspase-2-mediated transportation of caspase-3 is not well understood and requires further work. The localization of caspase-2 in the Golgi complex is implicated in the cleavage of its protein substrates such as golgin 160 and the subsequent fragmentation of this organelle during apoptosis, indicating its organelle-specific apoptotic functions [87,250,255]. However, the continuous compartmentalized trafficking of caspase-2 between ER and the Golgi demonstrated the physiological relevance of these compartments and appears to occur in response to ER stress signaling to execute the mitochondrial apoptotic pathway [28,132,274]. Additionally, Bax and Bak were also found to localize to ER and initiate a parallel pathway of caspase activation [275]. Recent studies have also highlighted the importance of caspase-2 activation in response to ER stress, indicating the importance of this protease in the mitochondrial apoptotic pathway or modulation of ER stress [163,276,277,278].

Additionally, it is worth noting that the identification of caspase-2-interacting adaptor molecules such as PIDD or RAIDD in these particular cellular compartments has expounded some of the mechanisms behind the activation of this enzyme. Nevertheless, the caspase-2 activation platform is generated in the cytosol, but not in the nucleus in response to DNA damage, heat shock, and cytoskeletal disruption [82,123,128]. Inspired by the notion that caspase-2 functions as an anti-apoptotic enzyme, the implication of a nuclear pool of caspase-2 upon its nuclear functions, such as tumor suppression and DNA repair, has also revealed its importance in nuclear functions of this enzyme along with its classical apoptotic role. It is unfortunate that despite caspase-2 having distinct features among other caspases and being implicated in many cellular functions, relatively few substrates of caspase-2 have been identified. Hence, to explore more about its role in physiological functions, the identification of novel caspase-2 substrates is indispensable.

### 4.1. Impact of Caspase-2 in Developmental Apoptosis

It is axiomatic that during development, extensive and selective destruction of unwanted or superfluous cells by apoptosis is a crucial physiological phenomenon. This process allows progression towards complexity in higher organisms and to achieve neuronal plasticity and tissue homeostasis for shaping the body [279,280]. On the other hand, PCD is also inevitable for the removal of unwanted cells in adult organisms to counterbalance the body’s proliferative requirements [280]. An in-situ analysis convincingly reported that caspase-2 is highly expressed in several types of embryonic tissues [15]. A relatively low and varied level of Nedd2 (caspase-2) mRNA was clearly detected in most parts of the adult brain, including the neurons of the cerebral cortex, pons, midbrain, cerebellum, and most other non-neuronal tissues such as lung, spleen, and kidney [15]. This study further indicated that those tissues with highly expressed Nedd2 have been subjected to high rates of PCD, suggesting that caspase-2 is an important mediator in developmental apoptotic machinery in mammals.

In contrast, recent studies have demonstrated that caspase-8 activity, but not caspase-2 or -9 is important for embryonic development [281], while caspase-2 mediated partial and transient apoptosis in mouse fetal oocytes [282,283]. Nevertheless, the activities of other caspases, including caspase-3, -6 and -9, have also been implicated in developmental apoptosis [4,15,284,285,286]. For example, during sensory neuronal development, caspase-3 is required for cell body apoptosis, while caspase-6 is required for axonal degeneration since naturally occurring neuronal cell death and axonal pruning are essential in concert to sculpt developmental neuronal connections [287,288]. It is noteworthy that during neuronal development, the timing of the death of various neuronal populations in different regions of the brain determines which distinct caspase orchestrates cell death of particular neuronal types from early embryonic to the early postnatal states, suggesting that certain caspases, including caspase-2, may take part in regional pruning to maintain the plasticity of the nervous system during development [17,289].

Recently, the inhibition of caspase-2 activities has been shown to significantly reduce experimentally induced neonatal brain damage in cerebral and white matter regions induced by hypoxia [213,290,291]. Another study also reported that in the embryonal cortex of neural-specific hypoxia-inducible factor-1α (HIF-1α)-deficient mice, vascular cells but not neural cells underwent oxidative stress-related, TNFR-mediated caspase-2-induced apoptosis, whereas the former underwent caspase-3- mediated cell death, indicating that two distinct types of cell death pathways are simultaneously triggered in the developing brain [289]. Recent studies using *Caenorhabditis elegans* (*C. elegans*), *Drosophila,* and mice have also emphasized that caspases are crucially important for dynamic cellular processes, including cell-fate determination, compensatory proliferation of neighboring cells and actin cytoskeleton reorganization in a context-dependent non-apoptotic function during development [281,292]. In summary, these results support the notion that overlapping and shared complex evolutionary activation mechanisms of caspases are involved in the regulation state and survival of an organism.

Despite being developmentally downregulated, caspase-2 gene expression can be reactivated in various adult neuronal cell types and other tissues [47,293] and that the processing of caspase-2 occurs rapidly in response to various intrinsic and extrinsic death stimuli in a variety of death paradigms [3,30,38,72,89]. Hence, it remains possible that caspase-2 is not a functionally redundant protease to play roles in trivial functions, rather it is speculated that caspase-2 is virtually a multifaceted caspase involved in the maintenance of the structural and functional integrity of an organism from its development and throughout life. In spite of the convergence of evidence, the discrepancy concerning the functions of caspase-2 in apoptotic pathways is still under much debate, rendering it more difficult to place its correct positioning in the apoptotic cascade [9,25,294,295]. Besides many of the unresolved controversies regarding its precise role in apoptotic function, several studies conducted over the past few years have emphasized convincingly that caspase-2 has a potential role in different apoptotic cell death paradigms [37,40,132,133,134].

The lack of an obvious phenotypical observation in caspase-2^-/-^ mice does not necessarily mean that this evolutionarily conserved enzyme is not involved in any crucial biological functions in mammals [19]. Interestingly, mice lacking other caspases, including caspase-1 and -11, have also failed to show an obvious phenotype, indicating that these types of caspases may be involved in developmentally induced regional pruning and the regulation of the plasticity of the nervous system [280,296]. Meanwhile, mice lacking caspase-6 show a pronounced hypoactive phenotype, learning deficits, age-dependent behavioral, and region-specific neuroanatomical changes associated with neurodegenerative disorders [297]. However, caspase-2^-/-^ mice develop normally and fail to show the explicit role of this protease especially in cell death of various types of tissues, suggesting that the enhanced or diminished apoptosis may result from the varied expression of pro- and anti-apoptotic isoforms of caspase-2 and dependent on the tissue of origin [17,298]. Another important issue relating to the lack of a phenotype in caspase-2^-/-^ mice may be the compensatory activation of other caspase family members in the absence of caspase-2 [8,35,87]. The compensatory changes in caspase activation are a general phenomenon for alternative cell death pathways, affecting the expression of an obvious phenotype, as have been detected in caspase-3 and caspase-9^-/-^ mice [299]. Compensatory mechanisms are not always in place for all death paradigms; for example, caspase-3^-/-^ mice show no evidence of compensatory activation of either caspase-6 or caspase-7 after ischemic insults [300].

In some developmental processes, highly redundant apoptotic mechanisms and complex degenerative processes are in place. For example, the degenerative cascade responsible for interdigital apoptosis in the developing avian limb involves the activation of caspases-2, -3, -6, -7 and -9, and the nuclear translocation of apoptosis-inducing factor (AIF) in a caspase-independent manner [4]. Hence, it is highly unlikely that developmental cell death is mainly caspase-2-dependent, and PCD can occur through other caspase-dependent or -independent manner depending on the cell context [11,280,301]. These observations clearly indicate that most developmental cell death can take place without the involvement of caspase-2. Overall, there is great complexity in cellular apoptosis and to determine which caspase can dominate the apoptotic pathway since the relative expression of both anti- and pro-apoptotic proteins determine the execution of specific apoptotic pathways.

### 4.2. Caspase-2-Mediated Cell Death in Response to Various Death Stimuli; Implications in Intrinsic and Extrinsic Pathways

Caspase-2 has been implicated in both mitochondria-dependent intrinsic and death receptor-mediated extrinsic cell death pathways, indicating that this protease has a critical role in cell death. Compelling evidence emphasizes that caspase-2 has the ability to function upstream or downstream in apoptotic signaling pathways (Figure 9) [7,24,65,69,259].

#### 4.2.1. Intrinsic Pathway

As an initiator, caspase-2 can act upstream of mitochondria either directly by the release of cytochrome c, AIF, and second mitochondria-derived activator of caspase protein (Smac) independent of Bid/other cytosolic factors or indirectly by the cleavage of Bid (Figure 9). Subsequently, Bax/Bak are generated, which lead to mitochondrial dysfunction through activation of caspase-9, and, eventually, the activation of executioner caspases and eventual cell death [83,116,251,302,303]. The physiological levels of purified caspase-2 were shown to cleave cytosolic protein Bid and the subsequent release of cytochrome c that is sufficient to activate the Apaf-caspase-9 apoptosome in vitro [30] and the ability of caspase-2 to cleave Bid is relatively higher than caspase-8 [251], indicating that caspase-2 is a direct effector of the mitochondrial apoptotic pathway.

Caspase-2 has been implicated in mitochondrial-mediated apoptosis in response to various death stimuli, including heat shock, serum deprivation, caspase, and RIP adaptor with death domain (CRADD), pore-forming toxin-mediated oxidative stress, and cytoskeletal disruption [30,37,39,133,251,304,305,306,307,308]. However, the implication of caspase-2 in response to heat shock is contradictory as caspase-9 is found as a critical initiator caspase activated in this paradigm and tBid may function to promote cytochrome c release during this process as part of a feed-forward amplification loop as downregulation of caspase-8, -2, or the caspase-2 adaptor protein RAIDD remained susceptible to heat-induced apoptosis in Jurkat cells [309]. Furthermore, the caspase-2 deficiency was shown to protect mouse embryonic fibroblasts upon heat-shock-induced stress [130,139], indicating a role in response to heat shock.

Caspase-2 is also required for stress-mediated apoptosis before MOMP, and both cytokine- and stress-induced cell death occurred through conceptually similar ways in which mitochondria act as an amplifier of caspase-2 activity rather than an initiator in this context [310]. Caspase-2 activation occurs downstream of apoptosome formation, dependent on both APAF-1 and caspase-9 upon certain death signals, including genotoxic stress [87,88]. ER stress-induced by drugs was found to associate with Bax/Bak and Bid-dependent apoptosis, but siRNA-mediated downregulation of caspase-2 failed to show significant effects under these conditions [132,311,312].

Although caspase-2 was initially identified as a neuronally expressed, developmentally downregulated caspase, it is required for neuronal death induced by various death stimuli including NGF deprivation, β-amyloid, and oxidative stress [40,41,47,304,313]. The processing of caspase-2 and trophic factor-deprivation induced cell death PC12 cells and sympathetic neurons occurred independently of caspase-3 [40]. Caspase-2 activation was redundant during seizure-induced neuronal death but suggested that parallel caspase pathways may circumvent deficits in caspase-2 function to complete the cell death process [314]. Despite this, caspase-2 has been shown to promote neuronal cell death upon trophic factor deprivation and after axotomy in retinal ganglion cells (RGC) and dorsal root ganglion neurons (DRGN) [134,135,136,137]. Hyperoxia-induced neurodegeneration triggered via the intrinsic mitochondrial pathway regulated key proteins, and the caspase-independent protein apoptosis-inducing factor, concomitant with the activation of caspase-2 and -3 in the neonatal brain has been reported [213].

It is worth noting that being a dimer in a solution like other effectors and the close sequence similarity of caspase-2 with caspase-9 raises a number of interesting questions whether this protease also possesses any unique structural features and functional similarities that of effectors and caspase-9 that enables it to undergo various distinct activation mechanisms [24,28]. For instance, caspase-2 is implicated as an initiator, upstream of mitochondria during certain stress-induced cell death pathways and recruited into a protein activating complex similar to APAF-1/caspase-9 apoptosome [54,80,147,315]. Under some circumstances, the initial caspase-2 activation takes place without its proteolytic cleavage, and its caspase-3 mediated processing is simply an amplification of caspase activation mechanism [52,54]. It is noteworthy that both caspase-2 and -9 have functional similarities in certain contexts. For example, NGF-deprivation induced cell death of caspase-2 null neurons depends on caspase-9 since neurons show a three-fold compensatory elevation of caspase-9 expression [316] and a double knock out caspase-2/caspase-9 reported that lymphocytes derived from these animals are normally sensitive to alternative cell death pathway [317]. Thus caspase-2 cannot bypass the apoptosome and depend on caspase-9 activation in some contexts.

As previously mentioned in this review, we certainly cannot exclude the possibility that the role of caspase-2 in genotoxic stress-induced apoptosis and caspase-2-meditated DNA damage-induced cell death is cell-type- and/or stimulus-dependent [80,147,156]. In support of this notion, Goniothalamin-induced oxidative stress, DNA damage, and apoptosis were shown to be caspase-2 and Bcl-2 in Jurkat T-cells [318], whereas 5-phenylselenyl- and 5-methylselenyl-methyl-2′-deoxyuridine-induced oxidative stress and DNA damage leading to apoptosis were caspase-2-dependent in cancer cells [122]. Moreover, it was reported that caspase-2 is important for MOMP in response to DNA-damage-induced insults [30,147,156,319], but in contrast, caspase-2 was found to be activated downstream of Bax and Bak and cannot bypass the apoptosome [87,320]. Taken together, it is not well understood whether caspase-2 functions as an initiator or effector in these paradigms.

#### 4.2.2. Extrinsic Pathway

Caspase-2 was also shown to associate with death-receptor-mediated apoptosis [29,77]. As already mentioned in this review, TRAIL-mediated apoptosis induced by caspase-2L has shown to be a promising therapeutic approach for tumor suppression. The implication of caspase-2 in death-receptor-mediated apoptosis is very efficient in tumor cells than normal wild type cells both in vivo and in vitro [321,322]. Some tumor cells are generally resistant to TRAIL-mediated cell death due to the presence of anti-apoptotic functions of specific proteins called CK2/PKCK2 [323,324]. Interestingly, a study showed that upon the downregulation of PKCK2, TRAIL-mediated apoptosis of tumor cells was orchestrated by caspase-2 via the processing caspase-8 (Figure 9) [91]. Furthermore, while caspase-2 deficiency was shown to protect hepatocytes from TNF-α mediated apoptosis [113], caspase-2 failed to protect embryonic fibroblasts in this paradigm [17]. These observations indicate that caspase-2-mediated cell death may also be tissue-specific.

In addition to TNFα, caspase-2 has also been implicated in FasL-mediated apoptosis in human leukemia Jurkat cells via the cleavage of its novel substrate cleaving sphingomyelin synthases-1 (SMS1) [325]. Likewise, caspase-2L isoform was also found to plays a role in the Fas-mediated cell death by contributing to caspase-8 activation at the DISC level in human leukemic variant cells [29]. Collectively, it appears that caspase-2 is implicated mostly in the intrinsic pathway of apoptosis in response to a variety of death stimuli, but is cell- or tissue-specific. Also, there are possibilities that it can function as an executioner caspase independently of the classical pathways, for example upon NGF-deprivation, Aβ-treatment, and Chk1 inhibition, when activities of other executioner caspases are neither sufficient nor necessary. The cell-death receptor-associated caspase-2-dependent cell death reflects its tumor suppressor function.

#### 4.2.3. Other Physiological Functions; Anti-Oxidant and Aging

Aging is associated with many factors, including oxidative stress, mitochondrial dysfunction, decreased glucose-6-phosphatase dehydrogenase (G6PDH), and dysregulation of caspase activities [326,327,328,329]. The progressive decline in physiological functions upon aging is directly influenced by the generation of reactive oxygen species (ROS) and is attenuated by a number of cellular sophisticated defense mechanisms, including DNA repair and antioxidant defense, to scavenge ROS. Surprisingly, caspase-2 associates with the relevant defense mechanisms; for example, in DNA repair, as described previously in this review [33,70,142], caspase-2S may act as a nuclear sensor to detect DSBs in response to genotoxic stress. Furthermore, the caspase-2 function is also found to be involved in the regulation of ROS. Caspase-2^-/-^ mice were shown to have impaired antioxidant defense, accumulation of ROS [142], and enhanced aging-related traits [330]. Not surprisingly, the age-dependent decline of G6PDH was found to provoke the caspase-2-mediated intrinsic pathway of cell death in age-related increase in muscle cell apoptosis in mice [90]. These observations indicate that the elimination of aged cells reduced the risk of tumorigenicity and thus maintained the genomic stability [19]. Moreover, the regulation of ROS was also orchestrated by p53 and implicated in nitric oxide-mediated apoptosis [331]. A novel antioxidant function for the tumor-suppressor gene p53 in RGCs has been observed, indicating p53 as a potential neuroprotectant for RGCs [332]. Based on these observations, it appears that p53 and caspase-2-mediated DNA repair and ROS scavenging are some of the distinct defense mechanisms in cells to determine its fate.

## 5. Conclusions

Increasing evidence indicates that caspase-2 can be activated via different mechanisms, but no such remarkable physiological evidence has been identified. Caspase-2 has unique features and regulates both pro-apoptotic and pro-survival functions, but these may depend on the cellular type and/or death stimuli and sate of a cell. The ability to modulate the life or death of a cell is recognized for its immense therapeutic potential. Complete genetic inhibition of caspase-2 may not seem beneficial and partial suppression may be advisable to achieve a benefit compared to risk. Encouragingly, emerging evidence now indicates that caspase-2 may contain a much wider context than predicted previously as an initiator caspase since it interacts with a myriad of molecules via numerous signaling pathways and an obvious gap in our knowledge concerns the mechanisms and functional significance of this. The engagement of caspases in apoptotic as well as non-apoptotic functions indicate that either pharmacological or genetic inhibition of caspases to prevent cellular death may have a great impact on the normal physiological functions of mammalian cells. Therefore, current and future research work must be directed towards vigilance of the broader physiological implications of caspases than initially thought. Hence, more efficient studies are still needed to be carried out to provide answers to many unresolved issues and to fix the missing pieces of the puzzle about caspase-2 activation and its significance on physiological functions and no doubt these studies will bring more surprising findings.

## 6. Perspectives

Notwithstanding a number of observations that indicate that the induced-proximity model undoubtedly leads to caspase-2 activation, there remains a number of open questions regarding the molecular mechanisms underlying the different activation pathways in different cell death paradigms. It is speculated that the formation of these specialized activating protein complex/platforms involves three major functions: (1) the sensing of cellular stress, damage, infection or inflammation; (2), multimerization of the platform; (3), recruitment and conformational activation of caspases. The more parsimonious explanation for the function of activating complexes in caspase-2 is that it may directly or indirectly influence caspase-2 activation. However, the forced oligomerization of initiator caspases under in vitro conditions may not recapitulate the physiological context in terms of protein expression levels, and the specific protein–protein interactions to induce conformational changes for their subsequent activation.

More importantly, it is not well understood whether caspase-2 activation by induced-proximity is always in place under physiological conditions. In this regard, we agree that the induced proximity model faithfully summarizes the process of initiator caspase activation at a general level but does not explain how the initiator caspases are activated at a mechanistic level under physiological conditions [94]. Nevertheless, the inhibition of either PIDD or RAIDD has profound effects on caspase-2-mediated apoptosis in certain cell types depending on death stimuli. In addition, the apoptotic signaling pathways in the same cellular system may differ depending on the initiating stimulus. Hence, our understanding of the molecular mechanisms underlying activation of caspase-2 is rather incomplete. Therefore, further structural, biochemical, and experimental validation studies are needed to address the many issues regarding the precise molecular mechanism for caspase-2 activation during different death paradigms.

The PIDD, RAIDD, and TRAIL mediated-caspase-2 activation pathways seem quite complex since a myriad of interacting molecules have been associated and implicated in both pro-survival and pro-apoptotic signaling. Collectively, it appears that the association of caspase-2 with these key regulatory molecules, especially with PIDD, occurs much earlier to provoke the signaling for pro-survival than apoptosis [135]. It also seems that caspase-2 mediated signaling pathways allow fine-tuning the different transcriptional activities to give a chance for cell survival rather than death and function as a death stimuli-regulated activator to determine the fate of certain cell types.

## 7. Future Directions

Despite the wealth of knowledge regarding the functions and activation mechanisms of caspase-2, there remain several unanswered questions that require further work:We know that the activation of caspase-2 is a two-stage process that yields a fully active caspase with enhanced catalytic-activity, but it is not clear whether processing/cleavage of caspase-2 by other caspases act as an amplification mechanism of death signaling events or represents a specific/cleavage mechanism (transactivation) where caspase-2 zymogens cannot be processed by auto-catalysis due to the existence of its several isoforms. To resolve this issue, further structural studies are required in caspase-2-mediated apoptotic pathways.Long prodomain caspases normally exist as monomers in solution at physiological concentrations, but caspase-2 is found as a dimer of the p19 and p12 subunits and the unique homo-dimerization of caspase-2 is driven by ligand binding and stabilized by a disulfide covalent bridge at the dimer interface. Even mutations of the relevant Cys residue at the dimeric interface did not affect the ability of recombinant caspase-2 to dimerize or to undergo autocatalytic cleavage, suggesting that inter-subunit cleavage may occur by an unknown intermolecular mechanism. Hence, further structural and biochemical studies are required to analyze the dimer interface stability both under in vitro and in vivo conditions.There are many studies that have indicated that caspase-2 exists as a dimer under physiological conditions, but why is caspase-2 activated through various distinct and complex activation mechanisms? This is the most compelling question about its mode of activation and needs further elucidation.We know that in the well-known proteolytic role of caspase-2 in apoptosis, both its association with activating platforms and consequent proteolytic cleavage is required; however, it should be clarified whether the non-apoptotic functions of caspase-2 in certain cell types require just its association with activating complexes but not cleavage.

## Figures and Tables

**Figure 1 cells-09-01259-f001:**
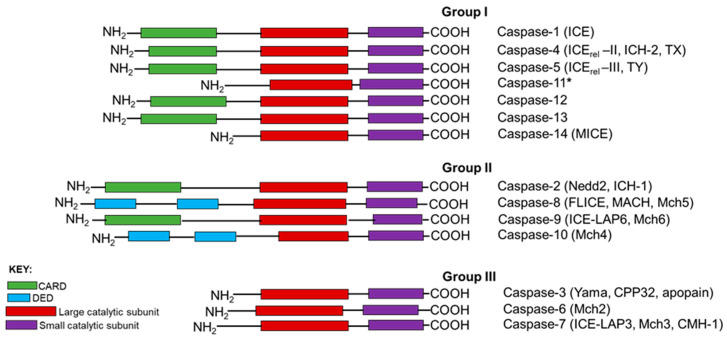
Structure of caspases. Three major groups of caspases are shown. Group I: inflammatory caspases; Group II: apoptosis initiator caspases; Group III: apoptosis effector caspases. The caspase recruitment domain (CARD), the death effector domain (DED), and the large (p20) and small (p10) catalytic subunits are indicated. * Caspase-11 is only found in mouse and is homologous to caspase-4.

**Figure 2 cells-09-01259-f002:**
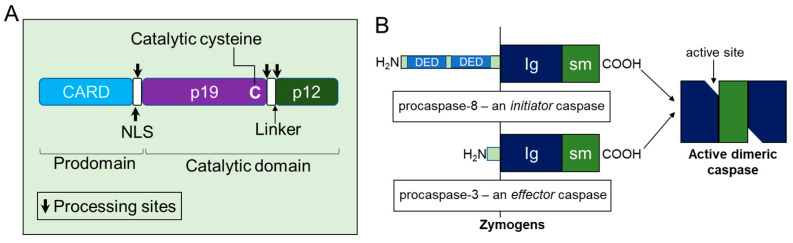
Structure of an initiator caspase. (**A**) Caspase-2, being an initiator caspase, contains an N-terminal CARD, followed by a large subunit containing the active site (p19) and a small subunit (p12). The structure also contains a nuclear localization signal (NLS). (**B**) Diagram showing the structures of the zymogens for caspase 8, another initiator caspase, and caspase 3, an effector caspase. The amino-terminal prodomains are shown in light green, large subunits are in blue, and small subunits are shown in green. The prodomain of caspase 8 contains two DEDs. Other initiator caspases may have a CARD instead. The heterotetrameric enzyme (right) results from proteolytic activation, as described in the text. Adapted from [7,45].

**Figure 3 cells-09-01259-f003:**
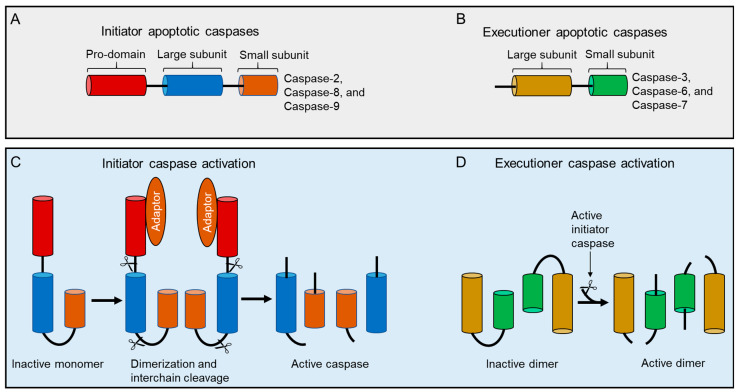
Mechanism of caspase activation by dimerization. (**A**) Initiator caspase (e.g., caspase-2, -8 and -9) organization: a prodomain precedes the catalytic domain, composed of two covalently linked (large and small) subunits. (**B**) Executioner caspase (e.g., caspase-3, -6 and -7) organization: a large subunit is covalently liked to a small subunit. (**C**) Initiator caspase activation: Initiators are inactive monomers carrying adaptor molecules (e.g., FADD and TRADD) that are activated by prodomain-mediated dimerization and interchain cleavage. (**D**) Executioner caspases are inactive dimers that are activated by cleavage of intersubunit linkers and by initiator caspases. Following activation, additional proteolytic events mature the caspases to more stable forms, prone to regulation.

**Figure 4 cells-09-01259-f004:**
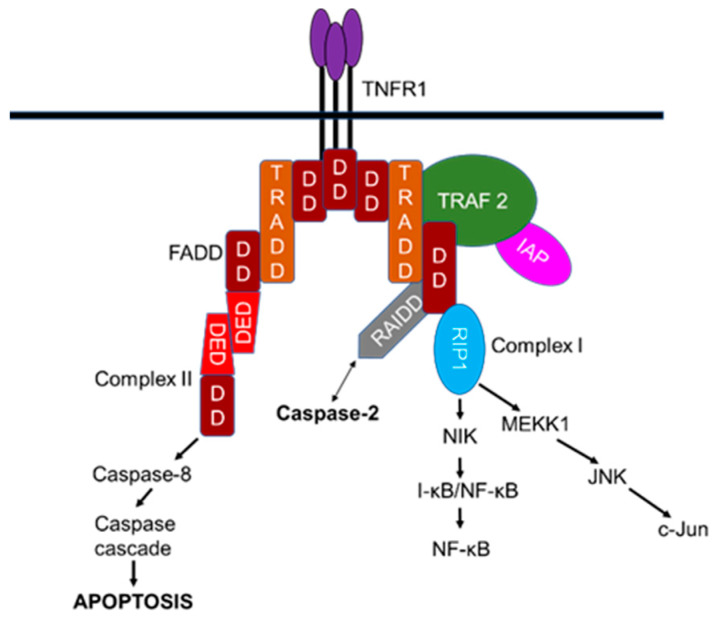
Signaling pathway of TNFR-1. Optimal signaling activity requires receptor trimerization. TNFR-1, with its death domain (DD), initially recruits TRADD (TNFR-associated death domain) and Fas-associated death domain (FADD), forming Complex II, which activates caspase-8 mediated apoptosis. TNFR-1 also induces apoptosis by activating caspase-2 through the recruitment of RIP (receptor-interacting protein), which also has a functional component that can initiate NF-κB and c-Jun activation, both favoring cell survival and proinflammatory functions via Complex I formation. DED, death effector domain; RAIDD, RIP-associated ICH-1-like protein with death domain; IAP, inhibitor of apoptosis proteins; MEKK1, mitogen-activated protein/Erk kinase kinase-1; JNK, c-Jun N-terminal kinase; NIK, NF-κB-inducing kinase; 1-κB/NF-κB, inactive complex of NF-κB that becomes activated when I-kB portion is cleaved.

**Figure 5 cells-09-01259-f005:**
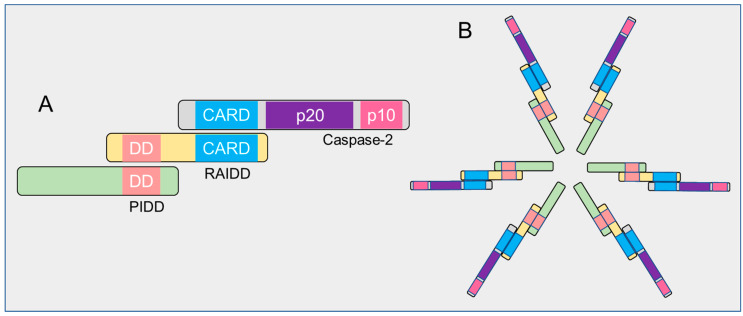
The PIDDosome. (**A**) Caspase-2 is linked via its CARD to receptor-interacting protein-associated Ich-1/Ced-3-homologue proteins with a death domain (RAIDD) which in turn associates with p53-induced proteins with a death domain (PIDD) via its death domain (DD). (**B**) A high molecular weight PIDDosome forms in response to genotoxic stress to mediate the apoptotic functions of caspase-2.

**Figure 6 cells-09-01259-f006:**
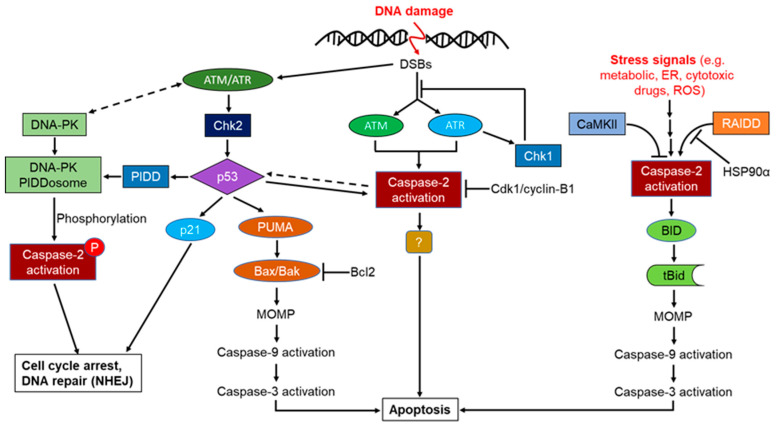
Functions of caspase-2 in DNA damage and cellular stress. After DNA double-strand breaks (DSBs), ataxia telangiectasia mutated (ATM) and ATM-related (ATR) kinases are activated and, in turn, phosphorylate and activate several target proteins, including checkpoint kinase 1 (Chk1) and Chk2. Chk2 activates the p53 response pathway, which can lead to cell cycle arrest. ATM/ATR activation leads to the activation of caspase-2 and apoptosis, following irreparable DNA damage by an unknown mechanism. ATR also activates Chk1, which can then act in a feedback loop to negatively regulate ATR and inhibit further activation of nuclear caspase-2. DNA-dependent protein kinase (DNA-PK) is also activated by DSBs, probably by activation of ATM/ATR, then forms a complex with p53-inducible protein with a death domain (PIDD) and caspase-2 (DNA-PK PIDDosome). This complex phosphorylates and activates caspase-2, which then initiates non-homologous end-joining (NHEJ) and DNA repair. Caspase-2 is also activated by other stress signals such as metabolic and endoplasmic reticulum (ER) stress, cytotoxic drugs, and reactive oxygen species (ROS)). Heat shock recruits RAIDD (receptor-interacting protein-associated ICH-1/CED-3 homologous protein with a death domain), which also activates caspase-2 (inhibited by HSP90α). Ca^2+^/calmodulin-dependent kinase II (CaMKII) inhibits caspase-2 activation and cell death in oocytes. Activated cytosolic caspase-2 cleaves Bid to its truncated form (tBid), which induces mitochondrial outer membrane permeability (MOMP), activation of caspase-9 and -3, and eventual apoptosis. P = phosphorous; PUMA = p53-upregulated mediator of apoptosis.

**Figure 7 cells-09-01259-f007:**
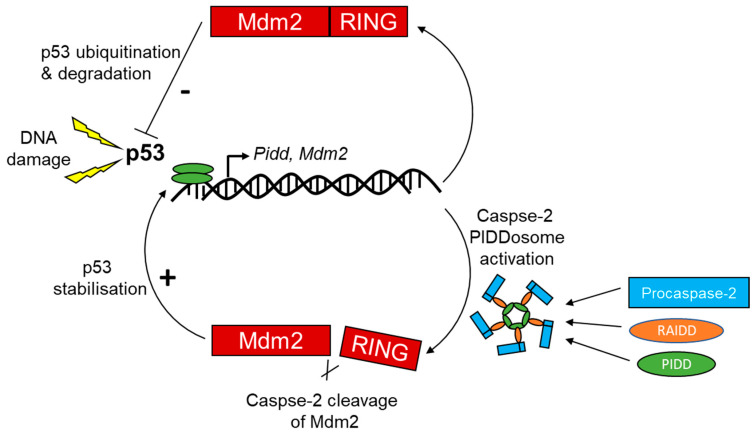
PIDD-mediated cell cycle arrest and drug resistance. PIDD-induced caspase-2 directly cleaves the E3 ubiquitin ligase Mdm2 at Asp 367, leading to loss of the C-terminal RING domain responsible for p53 ubiquitination. As a consequence, N-terminally truncated Mdm2 binds p53 and promotes its stability. Upon DNA damage, p53 induction of the Caspase-2-PIDDosome creates a positive (+) feedback loop that inhibits Mdm2 and reinforces p53 stability and activity, contributing to cell survival and drug resistance. These data establish Mdm2 as a cleavage target of Caspase-2 and provide insight into a mechanism of Mdm2 inhibition that impacts p53 dynamics upon genotoxic stress.

**Figure 8 cells-09-01259-f008:**
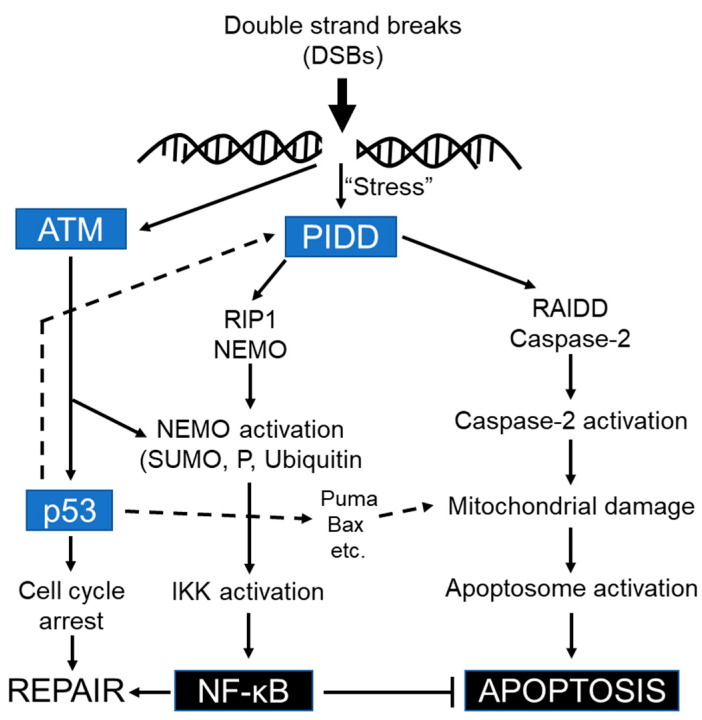
PIDD mediates NF-κB activation in response to DNA damage. Upon DNA damage, genotoxic stress is detected by PIDD, which causes its nuclear translocation and/or retention, allowing PIDD to accumulate in the nucleus and interact with RIP1 and NEMO, both of which are already present in the nucleus. NEMO sumoylation and RIP1 modification then ensue. In parallel, genotoxic stress can also activate protein kinase ATM, which phosphorylates sumoylated NEMO, leading to ubiquitination, nuclear export of NEMO, and subsequent IKK activation in the cytoplasm. ATM then cause cell-cycle arrest by activation of the p53 pathway and eventually repair of the damaged DNA. However, excessive DNA damage activates an apoptotic program to avoid possible cellular transformation. In addition to PIDD, DNA-damage-induced p53 transcriptional activity also leads to the induction of proapoptotic genes such as Bax, Puma, or Noxa. Transcription of PIDD and consequent activation of caspase-2, together with the activation of the other p53-dependent proapoptotic genes, eventually lead apoptosis. p53 can also indirectly activate PIDD and PIDD, thus acts as a molecular switch between survival or apoptosis, with the final outcome probably governed by the levels of damage.

**Figure 9 cells-09-01259-f009:**
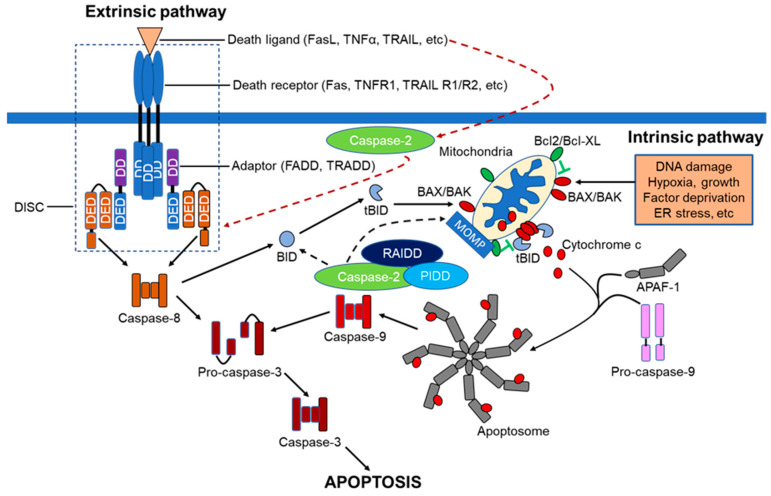
Extrinsic and intrinsic pathways of apoptosis. The extrinsic pathway is initiated by binding of death ligands (e.g., FAsL, TNFα, TRAIL) to their receptors (Fas, TNFR1, TRAIL), resulting in the formation of the death-inducing signaling complex (DISC), which contains the death receptor, an adaptor molecule (possessing a death domain (DD) and a death effector domain (DED)) and pro-caspase 8. Caspase-8 is autocatalytically activated and subsequently transmits the death signal to effector executioner caspases such as caspase-3, resulting in apoptosis. In the extrinsic pathway, bot death receptor and death ligands can recruit caspase-2-mediated activation of pro-caspase-8 to engage apoptosis (brown dashed lines). The intrinsic pathway is triggered by a number of factors, including DNA damage, hypoxia, growth factor deprivation, and ER stress. The death signal is sensed initially by the BH3-only protein, which then interacts with the downstream mediators of apoptosis (BAX and BAK). BAX and BAK undergo conformational changes, which lead to the formation of mitochondrial pores or increased permeability of the mitochondrial outer membrane, releasing apoptogenic compounds such as cytochrome c. Cytochrome c binds to APAF-1 along with pro-caspase-9 to form the apoptosome, which then activates caspase-9 and subsequent downstream executioner caspase such as caspase-3, -6, or -7, leading to apoptosis. Caspase-2 can activate BID or increase MOMP upstream of the intrinsic pathway of activation (black dashed lines). Abbreviations: TNF-α, tumor necrosis factor-α; TRAIL, TNF-related apoptosis-inducing ligand; FADD, Fas-associated death domain protein; TRADD, TNF receptor-associated death domain protein; BID, BH3 interacting-domain death agonist; tBID, truncated BID; BAX, Bcl-2 homologous antagonist/killer; BAK, Bcl-2 associated X protein; APAF-1, apoptotic protease activating factor-1; MOMP, mitochondrial outer membrane permeability.

**Table 1 cells-09-01259-t001:** Caspase-2 substrates identified to date.

Substrate	Physiological Function of Substrate	Effect of Cleavage	References
Bid	Apoptosis activator	Generation of tBid which interacts with BAX to promote MOMP	[30]
Casapse-2	Apoptosis	Activation	[30,55]
CERT	Sphingomyelin synthesis	Inhibition of sphingomyelin synthesis	[261]
eIFB4	Translation initiation	Unknown	[36]
Golgin 160	Vesicular trafficking	Apoptotic signalling?	[250]
HDAC4	Transcriptional corepressor	Altered subcellular localization	[262]
Huntingtin	CNS formation, anti-apoptotic, neuroprotective	Aggregation, cytotoxic	[263]
ICAD	Inhibition of CAD	Release from CAD, which subsequently fragments DNA	[264,265]
Mdm2	p53 degradation	Increased stability of p53	[266]
ΔNp63	Apoptosis	Abrogates transcriptional inhibition of TAp63	[267]
PARP	DNA damage sensing and repair	Inactivation	[268]
PKCδ	Cell cycle regulation and apoptosis	Activation, induction of apoptosis	[269]
Plakin	Part of adhesion junctions	Loss of cell-cell contact	[270]
ROCK1/ROCK2	Actin cytoskeleton organization	Cell detachment	[271,272]
αII-spectrin	Plasma membrane integrity	Cell shape disruption	[273]

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
