# Peer review of "The Role of Caspase-2 in Regulating Cell Fate"

_cells, 2020, doi:10.3390/cells9051259_

Round 1

Reviewer 1 Report

This review by Vigneswara and Ahmed is a comprehensive consideration of caspase-2 and physiology and pathophysiology, covering fundamental regulation to involvement in signalling and cell fate processes. The work is well written and cleanly presented. The figures were well-prepared and of high quality. I greatly enjoyed reading this review. It will be of interest to any biologist involved in cell fate and cell death research.

First, a few general comments:

  1. The authors clearly have a strength in writing. However, this review is massive due to the comprehensive nature – it’s effectively a crash course on everything there is to know about caspase-2. Unfortunately, it’s verbose to the point of muddying the reader’s comprehension. The content of this review spans a huge array of topics with very few figures, and those without existing expertise on caspase regulation or caspase-2 specifically may find it difficult to understand the concepts being described. The authors should review their manuscript and consider whether they are making their point in the most effective manner. I was particularly slowed down throughout Section 3 since there are very few figures and each sub-section refers to an entirely different pathway.
  2. Based on the title, I expected to get a clear message of “Can regulated caspase-2 be a determinant of cell fate?” I was then presented with the entirety of the literature explaining all the ways in which caspase-2 is involved in critical pathways which clearly are determinative of cell fate. Is the title effective? Is this really an open question? If so, it is not clear why. It seems very clear that caspase-2 as a regulator of cell fate has been recognized for decades, albeit with less popularity than caspase-3/7/9, -8 and -1. If this is indeed an open question, the authors should be more clear about (i) what are the open questions and (ii) what needs to be determined still. This may be done effectively in a summary/future directions paragraph towards the end, perhaps.

Some specific recommendations:

  1. Line 21 – do the authors possibly mean “physiological conditions” here? Pathophysiological conditions are discussed in the following sentences, and development and maintenance of homeostasis would not be considered “patho.”
  2. It would be appropriate the cite the original 1972 Kerr et all paper in the first sentence: doi: 10.1038/bjc.1972.33
  3. Figure 1 – synonyms are given next to each caspase name in parentheses, but the sub-identifier “murine” is specific for caspase-11. This is confusing. The authors might consider putting an asterisk next to caspase-11 and then explaining in the figure legend that caspase-11 is found only in mice and is most closely homologous to human caspase-4
  4. Figure 1- the synonym for caspase-2 should be (Nedd2, ICH-1), not just ICH-1.
  5. It may be helpful to explicitly state in the text that caspase-2 was also named ICH-1 in the early literature, in addition to Nedd2.
  6. Section 3, throughout, is extremely complex – more figures are strongly encouraged.
  7. Similarly, a unifying figure describing intrinsic-extrinsic activation pathways with caspase-2 (sections 4.2.1,2, and 3) is recommended.

Author Response

Thank you for your excellent and very helpful reviews. We are very grateful and hope that our changes have improved the article. We have answered all of your queries and have provided additional text and figures where requested.

1. The authors clearly have a strength in writing. However, this review is massive due to the comprehensive nature – it’s effectively a crash course on everything there is to know about caspase-2. Unfortunately, it’s verbose to the point of muddying the reader’s comprehension. The content of this review spans a huge array of topics with very few figures, and those without existing expertise on caspase regulation or caspase-2 specifically may find it difficult to understand the concepts being described. The authors should review their manuscript and consider whether they are making their point in the most effective manner. I was particularly slowed down throughout Section 3 since there are very few figures and each sub-section refers to an entirely different pathway.

Author response: This section is very complex due to the complexity of pathways and activation mechanisms involved in caspase-2-mediated cell death. However, we have added 2 new figures into Section 3 (Figure 4 and 6), removed some text and fewer subsections are now included. We hope it is much clearer now. We have also added Figure 9 in Section 4 to explain intrinsic and extrinsic apoptosis mechanisms. 

2. Based on the title, I expected to get a clear message of “Can regulated caspase-2 be a determinant of cell fate?” I was then presented with the entirety of the literature explaining all the ways in which caspase-2 is involved in critical pathways which clearly are determinative of cell fate. Is the title effective? Is this really an open question? If so, it is not clear why. It seems very clear that caspase-2 as a regulator of cell fate has been recognized for decades, albeit with less popularity than caspase-3/7/9, -8 and -1. If this is indeed an open question, the authors should be more clear about (i) what are the open questions and (ii) what needs to be determined still. This may be done effectively in a summary/future directions paragraph towards the end, perhaps.

Author response: We agree that this is not an open question and hence we have changed the title to: The role of caspase-2 in regulating cell fate.

Some specific recommendations:

1. Line 21 – do the authors possibly mean “physiological conditions” here? Pathophysiological conditions are discussed in the following sentences, and development and maintenance of homeostasis would not be considered “patho.”

Author response: The reviewer is right and we have amended this to take out ‘patho’, Line 23

2. It would be appropriate the cite the original 1972 Kerr et all paper in the first sentence: doi: 10.1038/bjc.1972.33

 Author response: Cited as reference 1, Line 23.

3. Figure 1 – synonyms are given next to each caspase name in parentheses, but the sub-identifier “murine” is specific for caspase-11. This is confusing. The authors might consider putting an asterisk next to caspase-11 and then explaining in the figure legend that caspase-11 is found only in mice and is most closely homologous to human caspase-4

Author response: We agree and this has now been changed as recommended. See Figure 1 and legend to Figure 1 (Line 51).

4. Figure 1- the synonym for caspase-2 should be (Nedd2, ICH-1), not just ICH-1.

Author response: Nedd2 added to Lines 58-59.

5. It may be helpful to explicitly state in the text that caspase-2 was also named ICH-1 in the early literature, in addition to Nedd2.

Author response: Now added in Lines 53-54.

6. Section 3, throughout, is extremely complex – more figures are strongly encouraged.

Author response: We have added 2 new figures (Figure 4 (Lines 323-334) and 6 (Lines 605-624) and have removed some text and in general tidied up this Section to make it more coherent.

7. Similarly, a unifying figure describing intrinsic-extrinsic activation pathways with caspase-2 (sections 4.2.1,2, and 3) is recommended.

 Author response: Added a new figure to link extrinsic and intrinsic pathways with caspase-2, now Figure 9, Lines 1241-1263.

Reviewer 2 Report

The manuscript by Vigneswara and Ahmed with title “Can regulated caspase-2 be a determinant of cell fate?” attempts to describe the current knowledge on the molecular and cellular mechanisms underlying the multifaceted process of caspase-2 activation. Also, the sub-cellular localization of caspase-2 and its crosstalk with factors were also discussed. This is topic is of high and emerging interest as the authors mentioned in their text too. I believe that this review’s topic will be certainly beneficial to the plant scientific community. Overall, the manuscript well written and well structured. Here below are some major concerns and questions:
1. This review is lacking reassessment of primary evidence and how this can be used to move this research field forward.
2. Nowadays, study about caspases and cell fate is a research hotspot. So, some caspases- and cell fate- related review articles have been published. For example, (1) Leist and Jäättelä, Nature Reviews Molecular Cell Biology, 2001, 2, 589-598; (2) Burgon and Megeney, Seminars in Cell & Developmental Biology, 2018, 82, 96-104; (3) Rohit et al., Current Pharmaceutical Design, 2018, 24, 3176-3183; (4) Kesavardhana et al., Annual Review of Immunology, 2020, 38. Thus, the difference between the manuscript and other published papers should be included in the introduction section. After that, the objective and novelty of the manuscript will be clearer.
3. “4. Sub-cellular localization of caspase-2”, but there are many discussions about the functions of caspase-2.
4. Figure 3 is artistic but lacks the necessary substance. Please redraw this figure.
5. Perspectives should be included in the conclusion section.

Author Response

  1. This review is lacking reassessment of primary evidence and how this can be used to move this research field forward. 

Author response: We have now provided a perspective section (Section 6, Lines 1363-1392) and a Future directions section (Section 7, Lines 1394-1418) to help in moving the research forward.   

  1. Nowadays, study about caspases and cell fate is a research hotspot. So, some caspases- and cell fate- related review articles have been published. For example, (1) Leist and Jäättelä, Nature Reviews Molecular Cell Biology, 2001, 2, 589-598; (2) Burgon and Megeney, Seminars in Cell & Developmental Biology, 2018, 82, 96-104; (3) Rohit et al., Current Pharmaceutical Design, 2018, 24, 3176-3183; (4) Kesavardhana et al., Annual Review of Immunology, 2020, 38. Thus, the difference between the manuscript and other published papers should be included in the introduction section. After that, the objective and novelty of the manuscript will be clearer.

Author response: All of these are excellent reviews but are very general. However, our review is primarily focused around caspae-2 and its interactions with the plethora of molecules so while we have referenced some of these articles, we have not explicitly stated the difference between these reviews compared to ours as our main focus is caspase-2. Instead, we have now added to the last paragraph in Section 1 (Lines 73-81) to tell the reader what we are setting out to do in this review. We then review the literature from the point of view of caspase-2 and compared and contrast what has been published regarding caspase-2. We believe this brings out the novelty of our review over the ones mentioned herein by the reviewer.  

  1. “4. Sub-cellular localization of caspase-2”, but there are many discussions about the functions of caspase-2.

Author response: We have removed some discussion (Lines 1083-1086) about functions of caspase-2, however, the function of caspase-2 appears to be specific to its localization and so not all of the discussion about function can be separated without losing meaning.

  1. Figure 3 is artistic but lacks the necessary substance. Please redraw this figure.

Author response: This figure is supposed to show the structure of the PIDDosome, which it clearly does, therefore we have not re-drawn this figure.

  1. Perspectives should be included in the conclusion section.

Author response: A perspective section is now included, Section 6, Lines 1363-1392.

Reviewer 3 Report

The manuscript under consideration is a review that summarizes available literature on the mechanisms of caspase 2 activation and its pro- and anti-apoptotic functions in different models. It considers a large array of studies and gives an overview of how the functions of caspase-2 can vary in cell-specific and context-dependent manner. The provided information is of interest; however, some points may be addressed to improve it:

  1. The text is full of mistypes and unclearly written fragments, which largely complicates reading and understanding. Proofreading and correcting mistakes would be highly beneficial for the manuscript. I will give few examples of this, but there are a lot of similar mistakes to be corrected:
    • “Consequently, both large and small cleaved 126 fragments (a heterodimer of a zymogen comprises newly cleaved small and large subunits) from two 127 procaspase molecules which are assembled to form active heterotetramers”. Verb is missing
    • “Enzymes that catalyze the phosphorylation of caspase-2 have been identified such as protein phosphatase-1 (PP1) and protein phosphatase-2A (PP2A) and these kinases have direct and indirect effects on caspase-2 dephosphorylation.” I guess here by “kinases” authors mean “phosphatases”. Otherwise, it is unclear about what they are talking about.
    • “This activating platform mediated phosphorylation appears to be indispensable for caspase-2 activation and its novel non-apoptotic functions in response to DSBs. Unlike PIDDosome, this NF-κB is thought to maintain G2/M cell cycle arrest and DNA repair regulated by the non-homologous end-joining pathway.” NF-kB was not mentioned previously in this section and mostly likely erroneously put into this sentence. But this is rather confounding and misleading.
    • “CD95-mediated DISC formation in response to DNA is also as an alternative PIDDosome-independent activation platform for caspase-2 [78].” I guess the authors missed the word “damage” after DNA here.
  2. Section 3 constitutes the largest part of the manuscript and contains the most interesting pieces of information. However, it is written unclearly and hard for comprehension. The section is split into many subsections, but there are large overlaps in information and it is not well organized. For example, proximity-induced dimerization is initially described in the very beginning of the section in “3.1. Caspase-recruitment activation complex-dependent proximity-induced dimerization and conformation models”, but suddenly reappears in the end in “3.10 DISC mediated activation”. I understand that there is a vast amount of information and it is hard to make it perfectly aligned, but perhaps it might be improved by introducing more clearly the key concepts in the beginning of the section and giving a brief description of what awaits ahead. In any case, better organization of information in this section would greatly improve the quality of the review and make it easier for an unprepared reader to understand.
  3. The title of the manuscript is “Can regulated caspase-2 be a determinant of cell fate?”. Though, the authors emphasize many times that caspase-2 can play very different roles depending on the cell type and specific context, they do not really give an answer to this question.

Author Response

1. The text is full of mistypes and unclearly written fragments, which largely complicates reading and understanding. Proofreading and correcting mistakes would be highly beneficial for the manuscript. I will give few examples of this, but there are a lot of similar mistakes to be corrected:

Author response: We have now read through the whole manuscript and have corrected small mistakes and typos throughout.

  • “Consequently, both large and small cleaved 126 fragments (a heterodimer of a zymogen comprises newly cleaved small and large subunits) from two 127 procaspase molecules which are assembled to form active heterotetramers”. Verb is missing

Author response: Amended, Line 141

  • “Enzymes that catalyze the phosphorylation of caspase-2 have been identified such as protein phosphatase-1 (PP1) and protein phosphatase-2A (PP2A) and these kinases have direct and indirect effects on caspase-2 dephosphorylation.” I guess here by “kinases” authors mean “phosphatases”. Otherwise, it is unclear about what they are talking about.

Author response: Amended, Line 735.

  • “This activating platform mediated phosphorylation appears to be indispensable for caspase-2 activation and its novel non-apoptotic functions in response to DSBs. Unlike PIDDosome, this NF-κB is thought to maintain G2/M cell cycle arrest and DNA repair regulated by the non-homologous end-joining pathway.” NF-kB was not mentioned previously in this section and mostly likely erroneously put into this sentence. But this is rather confounding and misleading.

Author response: Amened, Line 757.

  • “CD95-mediated DISC formation in response to DNA is also as an alternative PIDDosome-independent activation platform for caspase-2 [78].” I guess the authors missed the word “damage” after DNA here.

Author response: Amened, Line 902.

2. Section 3 constitutes the largest part of the manuscript and contains the most interesting pieces of information. However, it is written unclearly and hard for comprehension. The section is split into many subsections, but there are large overlaps in information and it is not well organized.

Author response: We have removed some of the text that was overlapping, improved the text with fewer subsections and more diagrams to make it clearer. This section is complex due to the complexity of functions that caspase-2 is involved in.

For example, proximity-induced dimerization is initially described in the very beginning of the section in “3.1. Caspase-recruitment activation complex-dependent proximity-induced dimerization and conformation models”, but suddenly reappears in the end in “3.10 DISC mediated activation”. I understand that there is a vast amount of information and it is hard to make it perfectly aligned, but perhaps it might be improved by introducing more clearly the key concepts in the beginning of the section and giving a brief description of what awaits ahead. In any case, better organization of information in this section would greatly improve the quality of the review and make it easier for an unprepared reader to understand.

Author response: We aligned the sections better by removing a large part of 3.10 DISC mediated activation. This includes deleting Lines 913-941.

3. The title of the manuscript is “Can regulated caspase-2 be a determinant of cell fate?”. Though, the authors emphasize many times that caspase-2 can play very different roles depending on the cell type and specific context, they do not really give an answer to this question.

 Author response: We agree and hence we have now changed the title to better reflect the subject of our review.